**communications** engineering

# Mars planetary insights and design framework for future in-situ aerial robotic missions
Vishal Youhanna ⬤ ✉, Leonard Felicetti ⬤ & Dmitry Ignatyev ⬤

This paper presents a comprehensive framework for designing and deploying aerial robots (aerobots) to revolutionise Mars exploration. The Martian environment, characterised by a tenuous atmosphere, extreme thermal variations, and diverse, often inaccessible terrain, presents fundamental challenges to the operational range and efficiency of conventional rovers and landers. Aerobots can overcome many of these limitations by enabling rapid regional surveys, accessing high-priority sites beyond rover reach, and supporting human exploration through environmental reconnaissance. Drawing on insights from past planetary missions, including the Ingenuity helicopter, the framework integrates planetary science constraints with aerospace engineering principles to address aerodynamic performance, structural integrity, autonomy, and environmental resilience. Central to this work is the Mars Aerobot Design Thinking Matrix, a decision-support tool that links mission objectives to testable engineering requirements, enabling systematic trade-off analysis across configuration, energy strategy, and operational margins. The proposed framework aims to guide the development of aerobots capable of sustained and scientifically productive operations in the unique conditions of Mars.

Mars exploration is a central scientific endeavour of the 21st century, motivated by questions about life beyond Earth and the planet's geological and climatic history. Decades of missions have revealed a dynamic past and a present-day environment characterised by a thin atmosphere, low temperatures, and frequent dust activity. Interplanetary missions carry high risk and demand technologies precisely adapted to celestial body's conditions. Success depends on tools that are both robust and tightly matched to the environment.

Landers and rovers have been highly successful but remain limited in traversing steep, complex, or distant terrains. As future missions target less accessible regions, autonomous aerial robots (aerobots) become important complements to surface assets. Aerobots can conduct rapid surveys, reach cliffs, canyons, and cave entrances, and cover large distances efficiently. They can also scout for future human exploration by characterising landing sites and environmental conditions. The first powered flight on another planet by NASA's Mars Helicopter (Ingenuity) in 2021, demonstrated the feasibility of aerial operations in Mars' thin atmosphere and opened the path to more capable platforms.

This work presents a comprehensive framework for the design, development, and deployment of aerobots in future Mars missions. It integrates insights from prior exploration with aerospace engineering considerations to address the environmental constraints that govern flight performance and reliability. The approach links aeronautics and planetary science, emphasising that design choices must be grounded in a detailed understanding of Mars, captured in the guiding metaphor that one should imagine breathing on Mars before planning to fly in its air[1]. The focus is on aligning concepts with the realities of atmospheric density, temperature extremes, dust, and topographic variability so designs are not only theoretically sound but also operationally viable.

Existing literature offers many innovative concepts but often gives limited attention to the practical challenges of in situ deployment. The framework here is intended to bridge that gap by organising environmental, topographic, and climatological drivers into a tractable set of design and operational decisions. It highlights the role of adaptive design, structural robustness, autonomous capability, and mission planning in sustaining flight under Martian conditions. A central output is the Mars Aerobot Design Thinking Matrix, a structured aid for balancing constraints and objectives across the mission lifecycle. The matrix supports decisions on configuration, energy strategy, autonomy, and deployment by making trade-offs explicit and tying them to environmental and operational margins.

Faculty of Engineering and Applied Sciences, Cranfield University, Cranfield, UK. ✉e-mail: vishal.youhanna@cranfield.ac.uk

https://doi.org/10.1038/s44172-026-00647-y     **Perspective**

The paper is organised as follows. First, it situates Mars exploration within the broader context of planetary missions, then outlines science objectives relevant to aerobot roles. Subsequent section examines how topography influences mechanical design and how climate—particularly atmospheric structure, temperature, pressure, winds, dust, radiation, and solar irradiance—affects performance and endurance. Practical aspects of design and delivery are then considered, including materials, autonomy, and deployment concepts. Finally, the Mars Aerobot Design Thinking Matrix is introduced as a unifying tool to maintain coherence between mission goals, environmental drivers, and testable system requirements.

## Comprehending planetary exploration development

Planetary exploration is a fundamental scientific vocation that advances understanding of the Solar System and addresses questions on origins, life beyond Earth, and Earth's place in the cosmos[2]. It drives technological innovation, inspires future scientists and engineers, and fosters international collaboration, while preparing for a sustainable human presence in space with broad societal benefits[2,3]. Appreciation of past and ongoing missions is essential because each has expanded knowledge of planetary environments, surfaces, and atmospheres and has shaped technologies that enable further exploration. Mars remains a primary target due to its potential past habitability, diverse water reservoirs, and parallels with early Earth. Motivation extends beyond scientific curiosity: Mars is a stepping stone for future human exploration and interplanetary travel. Designing aerial robotic systems depends on understanding Martian geology, climate, and atmospheric constraints.

This section reviews the historical evolution of planetary exploration, with emphasis on Mars–Venus comparisons and mission lessons. It then details Mars' exploration history, highlighting findings and technological advances that shape future aerial systems. These discussions reinforce the central objective of this work: developing an aerobot optimised for Martian conditions. By analysing exploration progress to date, the section establishes a clear link between prior missions and the advancements required to integrate aerobots into future planetary campaigns.

### Earth's two neighbours—first to most explored

Earth is surrounded by two adjacent planets, Venus and Mars. Although Venus is closer and often dubbed Earth's twin, Mars has attracted greater sustained exploration. However, Venus was the first to be examined closely by spacecraft[4]: NASA's Mariner 2 flew by in 1962[5], and the Soviet Venera programme[6] achieved the first successful planetary probe, landers, and lander–orbiter combinations. Surface operations on Venus are exceptionally short-lived; the record is 127 min (Venera 13, 1982), which also returned colour images and the first planetary audio Fig. 1a[7]. While no land rover has been sent to Venus, the Soviet Vega balloons (1985) remain the only atmospheric aerostats to operate on another planet[8], transmitting for 46 hours near 50 km altitude in Earth-like temperatures and pressures that could host extremophile microbes[9], sparking interest in future—but currently unplanned—Venus aerobot missions.

In shape and size, Venus looks like Earth, but they are the opposite twins of each other[10]. Being the hottest planet in the Solar System, Venus sustains an average surface temperature of 464 °C due to intense volcanic activity and a dense atmosphere rich in carbon dioxide and sulphuric acid, which generates atmospheric pressures up to 92 bar, creating an environment that crushes and corrodes spacecraft[9,11]. Besides the environmental challenges, future human settlements would experience drastic disorientation due to the planet's rotational and orbital properties: a single Venusian day is ~243 Earth days, longer than its ~225-day year; rotation is retrograde (Sun sets in the east), and the minimal axial tilt yields negligible seasonal variation[9]. Given these characteristics and challenges, planetary exploration preference has been shifted towards Earth's second-closest neighbour, Mars.

Planet Mars is about half the size and two-fifths the gravity with a total surface area equalling nearly all dry land of Earth. A Martian day (sol) is ~24.6 Earth hours, while local hours are 1/24th fraction of a sol and a year is

668.6 sols owing to the larger orbital radius[12]. According to the commonly used Mars calendar (Year 0 in 1953), the planet completed its 38th Martian year on 12 November 2024[13]. A Human on Mars can experience relief from the home-like four seasons, though longer and varied in length, caused by the planet's Earth-like tilted rotational axis. Temperatures vary widely, from 20 °C down to −153 °C, depending on location, season, and time of day. The current thin atmosphere is mostly Carbon dioxide, and the rest is nitrogen, argon, oxygen and water vapours[12].

Over four billion years ago, Mars lost its global magnetosphere, replacing it with an induced magnetosphere generated by solar-wind interactions at the upper atmosphere[14]. The resultant atmospheric loss contributed to surface desiccation and elevated radiation exposure. No microbial life as known on Earth has yet been detected in situ, but upcoming missions remain focused on biosignature detection. ESA's ExoMars mission (target 2028) will carry the Rosalind Franklin rover equipped with a drill capable of reaching 2 m depth to investigate geology and potential signs of past life[15]. Mars caves and lava tubes are natural shields against radiation and cold temperatures, that might be inhabited by Mars life and are potential habitats for future human explorers. The United States reaffirmed the goal of sending humans to Mars in the 2030s, stimulating public–private partnerships and research into new space-habitat technologies[16].

The exploration of Venus demonstrated the feasibility of long-duration atmospheric flight, as seen in the Vega balloon missions, despite the planet's extreme surface conditions. This contrasts with Mars, where its thin atmosphere poses significant aerodynamic challenges, requiring innovative lift solutions for sustained flight. Understanding these planetary differences is crucial for designing and developing aerobots optimised for Martian flight conditions, as discussed in the section "Design, Development and Delivery Considerations for Martian Aerobots".

### Standing on the shoulders of the giants—Mars exploration history

Mars is spotted from Earth as a reddish globe because oxidised iron dust is suspended in its atmosphere and lifted from the surface. Unlike Venus's opaque, thick clouds, Mars's atmosphere is <1% of Earth's sea-level density, giving orbiters a clear view for global mapping[11,12]. NASA's Mariner 4 (1964) made the first close approach and returned the first images of another planet from deep space[17], helping to overturn popular claims of intelligent life and irrigation "canals" inspired by Giovanni Schiaparelli's 1877 telescopic map, Fig. 1b[18]. In 1971, Mariner 9 became the first spacecraft to orbit another planet and produced detailed global maps[19,20], identifying two wonders of the solar system that would underpin future science priorities[21,22]: Olympus Mons, the Solar System's largest shield volcano (roughly three times Everest's height and ~100 times Mauna Loa's volume), and Valles Marineris, an ~8 km-deep canyon system about as wide as Australia[23].

Mars is also renowned for dust storms (as shown in Fig. 1c). When Mariner 9 arrived, it encountered a global storm so extensive the planet temporarily resembled Venus in appearance; only volcano summits protruded above the dust[18]. Mission controllers used reprogrammable software to delay surface imaging until the dust settled over the following months: an early demonstration of adaptable onboard and ground software as a mission enabler[18,20]. The Soviet Mars programme followed closely after Mariner 9 with Mars 2 and Mars 3, each carrying an orbiter and a descent module with a rover. Software inflexibility and limited fuel led to descent during the storm and mission loss at or shortly after entry; nonetheless, Mars 2 achieved the first impact on Mars and Mars 3 the first soft landing and a few seconds of image scan transmission[24]. The same principle translates to Martian aerobots: autonomous decision-making will be required to adjust flight paths and objectives to evolving environmental conditions.

Landing on Mars is hard because its thin atmosphere is unable to provide the cushioning effect to slow down a spacecraft upon its entry, as experienced on Earth or Venus. Moreover, it takes about 5 to 20 min for a one-way signal to reach Earth from Mars and vice versa, depending on the planet's orbital orientation[25]. Often deemed as 7 min of terror for mission operators—the time it takes for a spacecraft to enter, descent and land (EDL)

**Fig. 1 | Early planetary surface imagery and Mars environmental context. a** Colour picture of the surface of Venus from the Venera 13 Lander. Image credit: NASA/NSSDC (ID-VG00261,262). **b** Schiaparelli's 1877 surface map of Mars. Image credit: G. V. Schiaparelli (1877); reproduced from NASA NTRS (19750005657)[18]; public use permitted. **c** Before (left) and after (right) one month of the global storm in 2001 captured by NASA's Mars Global Surveyor orbiter. Image credit: NASA/JPL/MSSS (ID-PIA03170). **d** First Colour Image from Viking Lander 1 of the Mars ground. Image credit: NASA/JPL (ID-PIA00563). Panels (**a**), (**c**) and (**d**) are NASA material/public domain.

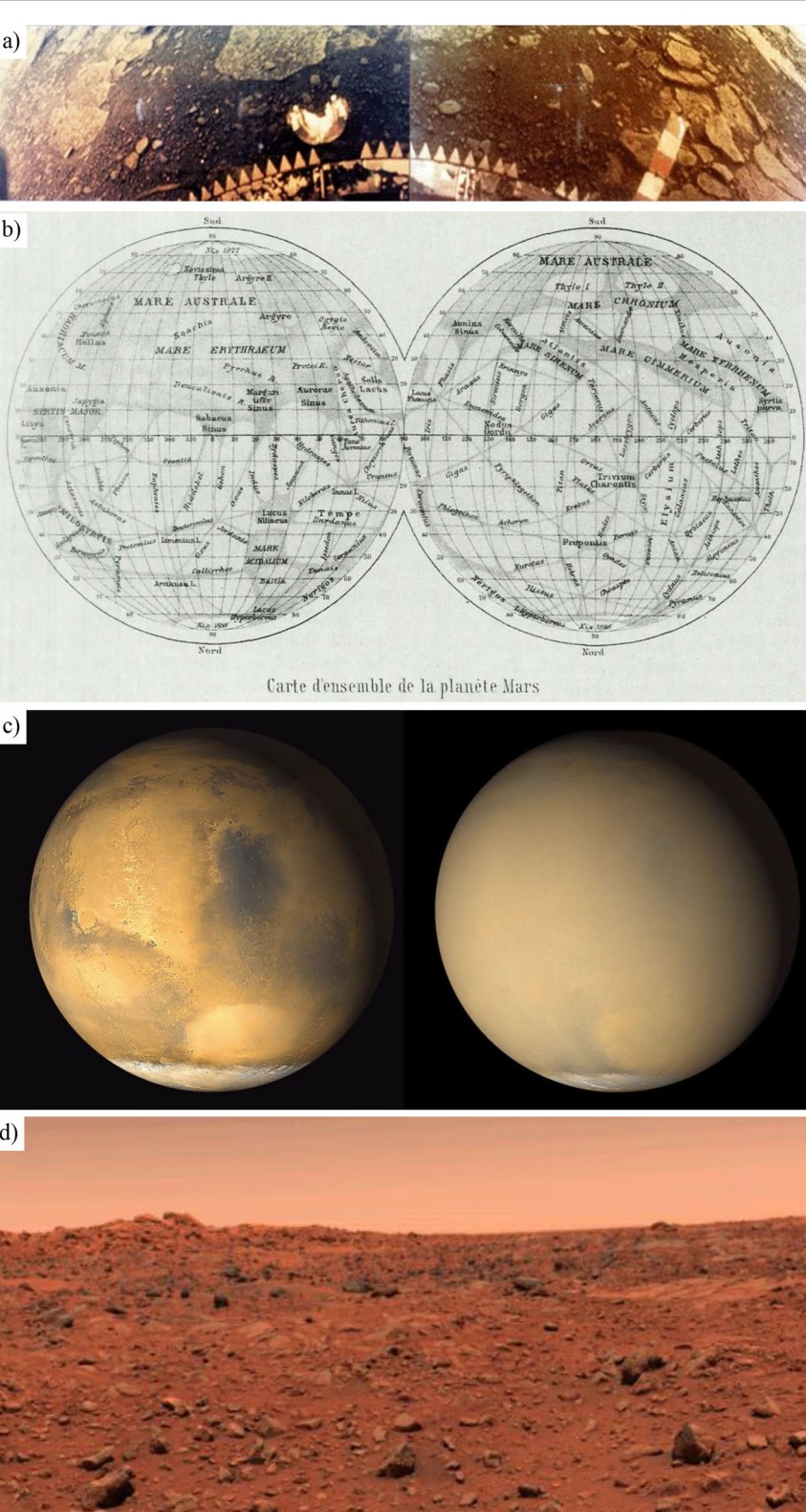

on Mars' surface from the top of its atmosphere, cannot be controlled in real-time. Along with mechanisms such as aerobraking, heat shields, parachutes and thrusters to make EDL successful, it is of utmost importance that the autonomous capacity of the control systems is well matched. Lessons learnt over decades of Mars missions, from both failures and successes, have advanced human expertise along with evolving technology to the point that, in early 2021, China National Space Administration (CNSA) deployed

Tianwen-1—the first mission ever to deliver an orbiter, lander, and rover on a nation's first attempt at Mars[26].

The Martian environment poses challenges to the alien exploratory bodies, unlike the hostile ones of Earth's twin, which have remained practically manageable for engineers, enabling more successful exploration of Mars. Viking 1, the first truly successful Mars lander (1975), was designed to operate for just 90 Earth days but lasted over six years, Fig. 1d[27]. The record

was surpassed by Opportunity (2004–2018), which travelled 45 km before a global dust storm disabled its solar panels[28]. Such extended lifespans have led NASA to expand the primary operational duration of their recent Mars missions from days to multiple Earth years, while remaining conservative with an allocation of about 30 days for the primary phase for technology demonstrators such as the rover Sojourner (1997)[29] and the helicopter Ingenuity (Mars Mission 2020). However, the helicopter transitioned from the technology demonstration phase to the operational phase, to aid its mission companion Perseverance—the most advanced Martian rover with a top speed of 0.04 m s$^{-1}$ on hard flat ground[30], that travelled around 40 km by 2025 since its landing in February 2021[31]. Being the predecessor of Perseverance in terms of design, Mars Science Laboratory's (MSL) Curiosity rover, comparatively, has covered a distance of around 36 km in its career of about 13 years on Mars[32]. Relatively, Martian spacecraft are slow by Earth transport standards but have been astonishingly successful as planetary explorers.

The historical evolution of planetary exploration underscores the growing need for high-mobility systems capable of overcoming terrain and atmospheric challenges. Ingenuity has become a step forward in increasing the pace of Mars exploration via aerial survey. Since landing, it completed 72 short autonomous pre-commanded flights before being damaged in January 2024, accumulating approximately 129 min of flight time, covering 17 km, reaching altitudes of up to 24 m, and achieving ground speeds of up to 10 m s$^{-1}$ [33]. This pace can be further increased with the next generation of aerobot designs. The following sections introduce a structured design framework that systematically addresses these constraints, ensuring that future aerobots are optimized for sustained Martian flight.

### Mars exploration science goals and requirements

Past, present, and future Mars missions have been guided by the NASA's Mars Exploration Program Analysis Group (MEPAG) report, first published in 2001 and updated biennially. This consentaneous statement of the Mars scientific community defines four major scientific goals: (1) Life, (2) Climate, (3) Geology, and (4) Preparation for Human Exploration[34]. Liquid water is a necessary prerequisite to life on Earth, therefore "following the water" traces remains central to biosignature and habitability investigations. Notably, the objective of characterising the atmosphere for safe robotic and human spacecraft flight first appeared under the Climate goal (MEPAG 2005) and was later moved under Human Exploration as an engineering application[35]. Ingenuity Helicopter's in situ demonstrations have advanced understanding of Martian aircraft flight characteristics.

Mars features various potential mission zones with distinct territorial characteristics such as plains, highlands, polar regions, underground caves, steep hills, and deep valleys—each demanding customised aerobot designs as discussed in Youhanna et al. [36]. Landing site selection begins with scientific promise but must first meet engineering and EDL safety requirements—such as latitude, elevation, terrain slopes, rock height, wind limits, and surface bearing strength—before prioritising sites of high scientific value aligned with MEPAG's four goals[37]. The Mars 2020 mission illustrates this process, with Jezero Crater chosen from 55 candidates for its astrobiological significance while meeting stringent engineering criteria. Historically, the selection team has conservatively opted for plains to simplify landing and rover traversal; for instance, Valles Marineris was shortlisted for both Curiosity 2012 and Perseverance 2020 rovers but rejected due to rugged terrain and long traverses to science targets[38]. Other notable Mars sites over the past decade, including Utopia Planitia (Tianwen 1 Mission 2020), Elysium Planitia (InSight Mission 2018), and Gale Crater (MSL Mission 2012), share similarities in surface flatness and roughness with Chryse Planitia (+22°, 312° E), the lander Viking 1's site in 1975 (Fig. 1d)[19]. Although the 2020 mission introduced Range Trigger and Terrain Relative Navigation for precision landings in rougher terrains[39], near-term missions are still expected to prefer flatter sites until high-resolution mapping of riskier but scientifically desirable regions is complete. Consequently, aerobot endurance and speed become pivotal to bridge distances from safe landing ellipses, with next-generation drones expected to produce high-resolution

maps of previously unmapped terrains[36]; where aerobots are included, some site-selection conservatism can be eased[38].

## Martian aerobot science missions subject to areography

The red planet has a diverse landscape with local environmental variations that offers numerous opportunities for future in situ missions but also demands familiarity with its areography, a specific term for Mars physical geography, when designing a mission-specific aerobot. The success of such systems relies on their ability to adapt to the planet's unique topography, climate, and environmental conditions, which directly shapes mission design, operational feasibility, and scientific return. This section identifies constraints and opportunities across Mars' diverse terrains and motivates mission-specific adaptations to navigate extreme landscapes efficiently.

### Mars topography influencing aerobot mechanical design

Reusable Martian aerobots, like Earth aircraft, will spend a significant amount of time on the ground being stationed, recharging their power source, conducting science or moving via dedicated landing gear. Mars' topography, from vast plains to towering volcanoes and deep canyons, presents distinct mechanical and operational challenges for aerobots. This subsection examines how elevation differences, surface roughness, and geological features dictate design choices, including structural sizing, resilience, and obstacle avoidance systems, ensuring aerobots can safely traverse and explore the varied landscapes of the planet.

**Mars global topography**. Mars global topography has a sharp contrast feature known as Martian Dichotomy, which divides the lowlands of the northern hemisphere and the highlands of the southern hemisphere with an area ratio of roughly 1 to 2, respectively, as mapped in Fig. 2[40]. Planetary surface ages are grouped into Noachian (3.9–3.5 Ga), Hesperian (3.5–1.8 Ga), and Amazonian (1.8 Ga–present), with the southern highlands preserving early impact records, whereas northern lowlands show resurfacing by volcanic flows[41]. Mars' zero-elevation datum (areoid), analogous to Earth's geoid, broadly traces the outlines of hypothesised ancient oceanic basins from periods when the planet had a thicker atmosphere[42]. Notably, all successful missions have landed in these low-elevation regions, targeting evidence of past life.

A meridional slice at 0° longitude shows an average north–south elevation difference of ~6 km, equivalent to a ~0.036° slope (Fig. 3a)[43]. The United States Geological Survey (USGS) has divided the surface into 30 Mars Chart (MC) quadrangles for standard cartography (Fig. 3b)[44,45]. The major topographic and gravitational anomaly that covers about one-fourth of the planet's surface is the Tharsis bulge or rise (about 170° W to 40° W, −50° to 50°), which elevates up to 10 km above the surrounding terrain, excluding the giant volcanoes it embeds[46].

These attributes such as datum, elevation gradients, and regional rises, affect atmospheric pressure, density and temperature and thus aerobot performance, flight efficiency, and landing feasibility. Transition zones between dichotomy provinces are also scientifically valuable and influence planning. In practice, global topography provides the first-order map for identifying viable operating altitudes, selecting low-elevation sites favourable for EDL, and anticipating energy requirements for flights across steep terrain.

**Mars orders of relief**. A five-level "orders of relief" scheme (by analogy with Earth) provides a hierarchical mental map of Martian landscapes[41]. The first order expresses the global dichotomy, while the third reflects geologic time. The second, fourth, and fifth orders progress from large planetary features to regional and local elements—aspects of this hierarchical classification are shown in Fig. 4(Top)—which guides both mission-specific aerobot design and the matching of pre-designed vehicles to appropriate sites. For aerobot design, the fourth-order regional features are key drivers of vehicle architecture and of mechanical and operational requirements, including propulsion sizing to handle

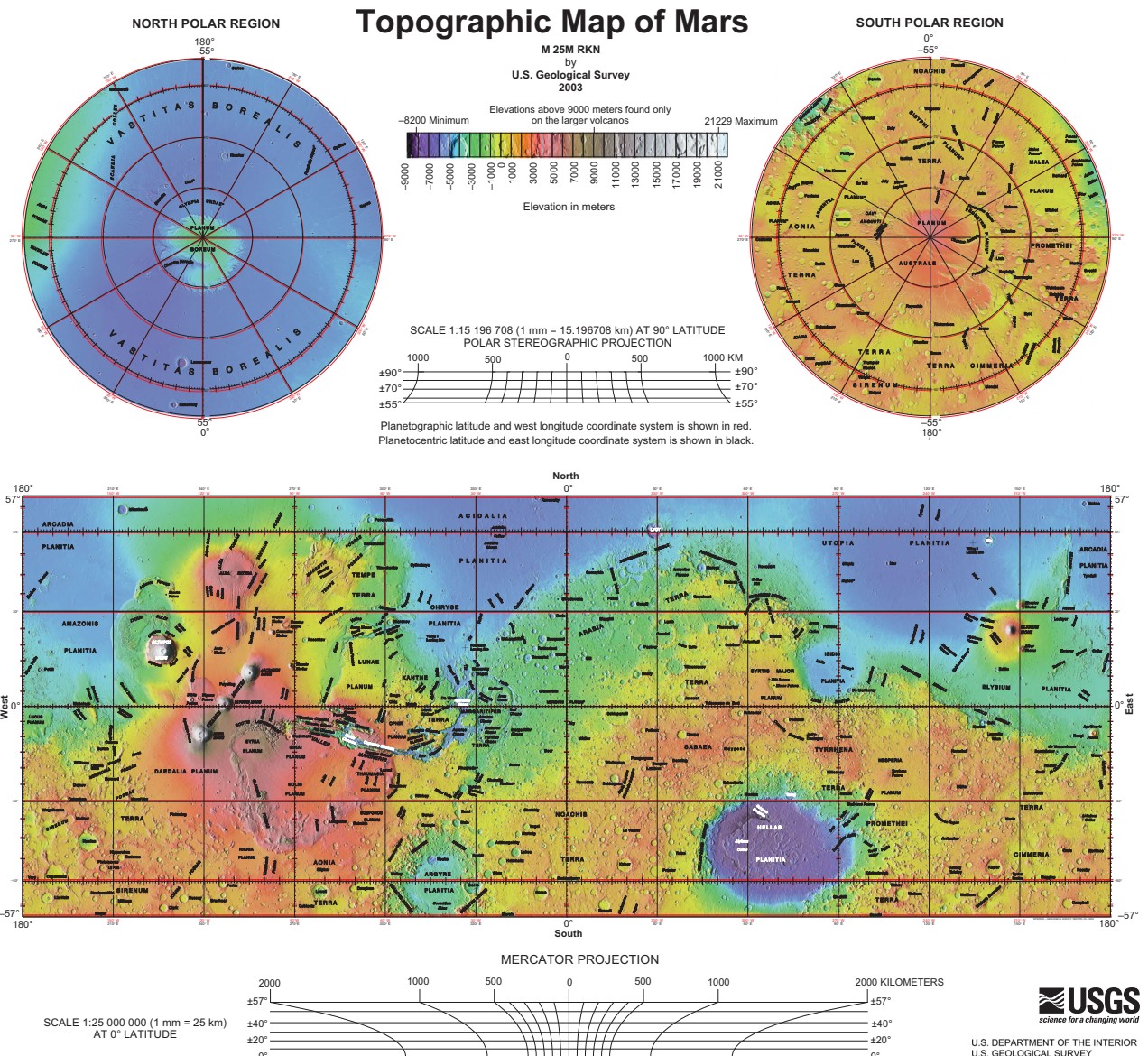

**Fig. 2 | Global topographic map of Mars and aerographic reference framework.** Topographic Map of Mars (M 25 M RKN; USGS, 2003), based on data from the Mars Orbiter Laser Altimeter (MOLA) an instrument on NASA's Mars Global Surveyor (MGS)[40]. Aerographic grid adopts a prime meridian through Airy-0 within crater Airy[43]. Map modified for clarity. USGS public domain.

elevation changes, structural and control authority for rugged terrain, and hazard avoidance for complex relief. Fifth-order local roughness such as rocks, boulders, small gullies and soft deposits then refines these requirements at the site level, influencing undercarriage design, ground clearance and autonomous landing capability, especially where coordination with a rover demands precise touchdowns. Figure 4 (Bottom)[47,48] further contextualises this relief classification by depicting real Martian topographic features.

On Mars, diverse terrains exhibit multiple 4th-order relief features. One example is the fretted terrain along the Martian Dichotomy, marking a sharp transition between lowlands and highlands with 1–2 km cliffs rising from smooth plains dotted with mesas, buttes, and yardangs[49]. Another distinctive features, the *Medusae Fossae* Formation (MFF) covering part of Aeolis Mensae, is a soft, layered volcanic ash deposit wrapped with yardangs, ridges, and grooves, and identified as the planet's largest single dust source[50]. The chaotic terrain near Valles Marineris outflow channels (Fig. 4i), with its jumbled blocks and fractures from ancient floods, poses extreme exploration challenges. Also notable are pit craters, or skylight caves (Fig. 4e), in volcanic

provinces that open into unexplored subsurface lava tubes, requiring bespoke aerobot designs. The HiRISE programme provides publicly accessible, high-resolution imaging since 2006[47], enabling Earth analogue comparisons to simulate mission scenarios. Overall, the orders-of-relief framework links geomorphic context to aerobot propulsion, control, hazard sensing, and mapping strategy, while underscoring surface asset synergy and deployment from validated EDL zones to mitigate terrain-driven risks. The Martian atmosphere, shaped by both global and regional climate, remains the other key driver in defining aerobot aerodynamic design.

**Mars gravity anomalies.** Mars' average surface gravity is 3.73 m s$^{-2}$ [51] (~38% of Earth) but varies due to uneven topography and internal structure. NASA's Goddard Mars Model-3 (GMM-3) free-air gravity map (120 km resolution; Fig. 5a) charts these anomalies in milliGals (1 Gal = 0.01 m s$^{-2}$), with most within ±500 mGal and higher values concentrated in major volcanic regions[52,53]. These variations from the mean gravitational acceleration, less than about 1%, produce only a small change in effective weight compared with the larger variability

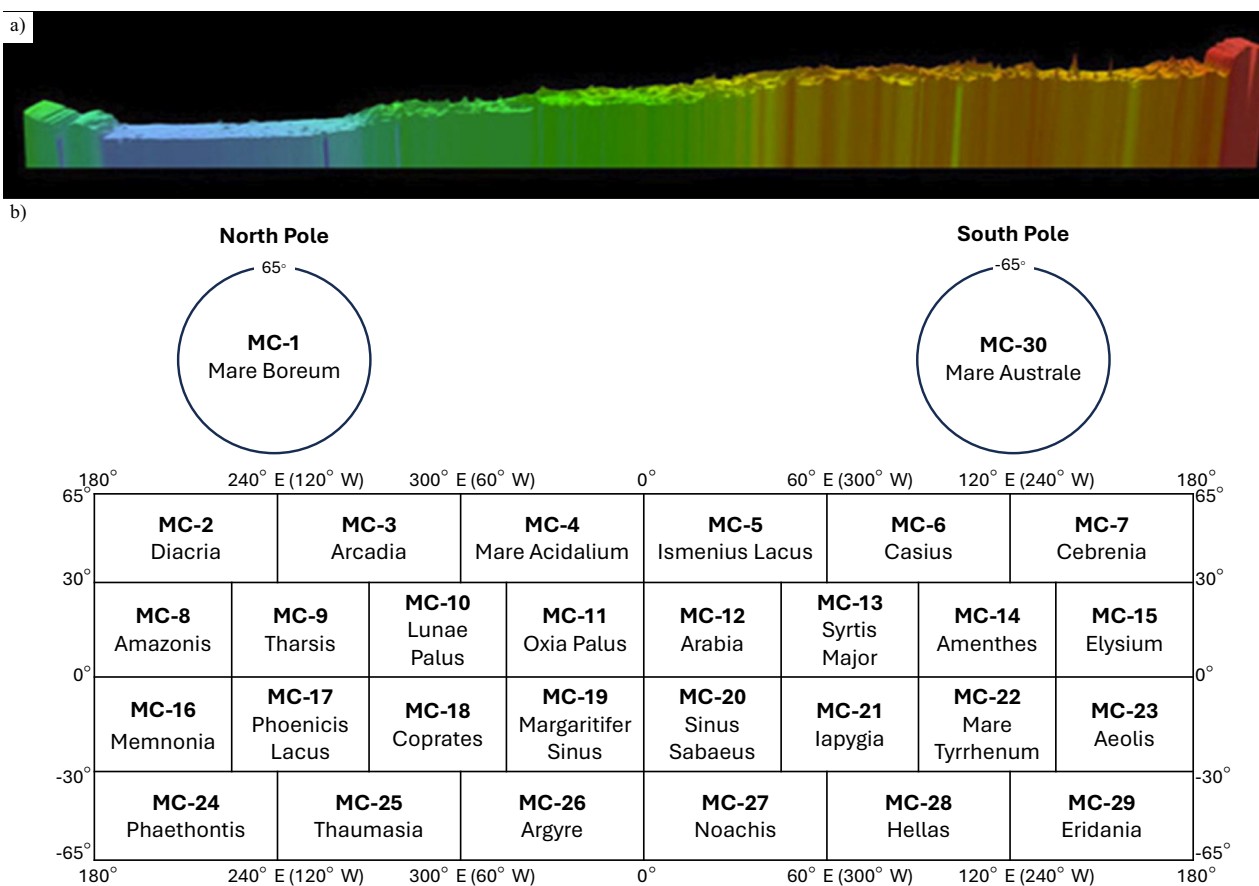

**Fig. 3 | Mars global referencing and cartographic context for aerobot mission planning. a** Slice runs showing a slope of 0.036° from the north pole (left) to the south pole (right) at 0° Longitude. Image credit: NASA/MOLA Science Team (public domain). **b** Mars Cartographic Chart. Credit: USGS Astrogeology Science Center (public domain); Redrawn based on source[45].

introduced by atmospheric density, winds and dust accumulation on the vehicle, and are therefore a secondary consideration for most low-altitude aerobots. Location-specific gravity values can still be useful to refine weight estimates, lift margins and instrument calibration in cases where accurate local gravity is required for performance or control analysis, particularly when vehicles operate close to such limits, and to support mission-level planning for long-range concepts such as fixed-wing gliders or hopping vehicles that travel between sites via ballistic flights.

At mission level, such fractional differences in gravitational acceleration are relevant for high-fidelity trajectory and energy modelling, particularly for orbit insertion and EDL, where experience from past missions has shown that simplified gravity assumptions contributed to landing-condition errors[54]. The GMM-3 field primarily provides regional geophysical context: in combination with MOLA topography and orbital imaging, it helps delineate major basins, volcanic provinces and crustal-thickness contrasts, supporting broad landing-site assessment and more accurate orbit and descent-trajectory predictions for spacecraft[52]. In the longer term, sustained aerobot campaigns could also provide low-altitude gravity measurements, complementing orbital models and improving the characterisation of subsurface structure relevant to future landing and exploration missions.

## Mars climatology influencing aerobots performance

The Martian climate, characterised by a thin atmosphere, extreme temperature fluctuations, and frequent dust storms, profoundly impacts aerobot performance. This subsection analyses how atmospheric density, wind patterns, solar irradiance, and radiation levels influence power requirements, flight stability, and operational endurance, underscoring the need for robust environmental adaptability in mission planning.

**Orbital mechanics driving Martian environment.** Mars' orbital and axial properties shape its climate and atmosphere. With an obliquity similar to Earth's, it experiences four seasons across 12 Martian months each spanning 30° of solar longitude (Ls) and ranging from 46 to 67 sols, beginning at Ls = 0° (northern spring), as illustrated in Fig. 5b[55]. Mars' higher orbital eccentricity (~0.1 vs Earth's ~0.02[51]) causes larger insolation variations between perihelion and aphelion, producing hemispheric asymmetries: southern spring (142 sols) and summer (154 sols) are shorter and more intense; northern spring (194 sols) and summer (178 sols) are longer and milder[41]. These orbital effects modulate temperature, pressure, winds, and dust activity through the year, directly influencing lift, stability, energy efficiency, and navigation strategies for aerobots.

Mars' surface temperature averages −73 °C[56], ranging from 20 °C to −153 °C[12], with the coldest conditions in polar winters and the warmest in southern subtropics near perihelion. Tropical regions can experience diurnal swings exceeding 100 °C[56]. Surface atmospheric pressure varies strongly with elevation—from 14 mbar in Hellas Planitia to 0.7 mbar atop Olympus Mons[57]—averaging about 6.1 mbar at the areoid datum[42], roughly 0.6% of Earth's sea-level pressure. This pressure fluctuates by up to 60% annually, compared to 20% on Earth[41], due to semi-annual polar cap variations, with deeper minima in southern winter. Mars' orbital eccentricity drives this asymmetry: during southern winter, $CO_2$ (95% of the atmosphere[51]) condenses onto the south polar cap below −123 °C, lowering global pressure by 25–30%[57], then sublimates in summer to raise it to its peak. Polar caps persist year-round but differ in composition: summer sublimation at the north pole exposes a water-ice cap, while the south pole retains $CO_2$ frost[56]. Winter clouds of water and $CO_2$ ice form over both

**2nd ORDER - Planetary Features**

- **Great Craters**
  - 4 great craters: Hellas Planitia, Argyre Planitia, Isidis Planitia, Utopia Planitia
  - Size range ~1500 - 3300 km in diameter (D)
- **Giant Volcanic Provinces**
  - Tharsis Rise (including surrounding region) (Fig 4. c)
    - Centered at 0°Lat, 245° E Long., size range D ~8000 km, rise upto 7 km
    - 5 Gigantic Volcanic Mountains: Olympus Mons (~21km tall), Alba Patera, Arsia Mons, Pavonis Mons, and Ascræus Mons
  - Elysium Rise (Fig 4. d)
    - Centered at +25° Lat, 146° E Long., size range D ~2000 km, rise upto 5 km
    - 1 Gigantic Volcanic Mountain and 2 Tholi: Elysium Mons (~13km tall), Albor Tholus and Hecates Tholus
- **Rift Zone**
  - Valles Marineris - a system of canyons
    - Below 0°Lat, Right to Tharsis, Length ~4000 km, Width ~700 km, local Depth ~ 7-10 km
- **The Polar Ice Caps**
  - North Pole Ice Cap - seasonal CO2 ice toping
    - D ~1000 km, ~1 km below areoid datum
  - South Pole Ice Cap - permanent CO2 ice topping
    - D ~350 km, ~5 km above areoid datum
- **Syrtis Major**
  - Dark triangular giant patch - telescopically conspicuous because of dark albedo of basalt
  - Above 0°Lat, around 60° E Long.

**4th ORDER - Regional Features**

- **Impact Features**
  - Craters of all sizes, at every level of superposition and modifications (Fig 4. a,b, k)
- **Tectonic Features**
  - Large volcanoes mons (Fig 4. c,d)
  - Small volacanoes and tholi (Fig 4. d)
  - Lava tubes and pit caves (Fig 4. e)
- **Fluvial Features**
  - Deltas (Fig 4. f)
  - Valles (Fig 4. g)
  - Gullies (Fig 4. h)
- **Mass Wasting**
  - Landslides and avalanches (Fig 4. h)
  - Chaotic terrains (Fig 4. i)
- **Aeolian processed features**
  - Mesas and buttes (Fig 4. j)
  - Yardangs (Fig 4. j)
  - Dunes (Fig 4. a, j, k, m)
  - Softened or inverted craters (Fig 4. k)
- **Glacial Features**
  - Polar Glaciers
  - Crater Glaciers (Fig 4. l)
  - Periglacial patterns - swiss cheese, brain terrain, glacial geysers (Fig 4. m, n, o)

**5th ORDER - Local Features**

- **Boulders, Rocks and Bluberries Haematite**
- **Dust Devil Tracks and Dust Streaks**
- **Frost Deposits**
- **New Gullies**

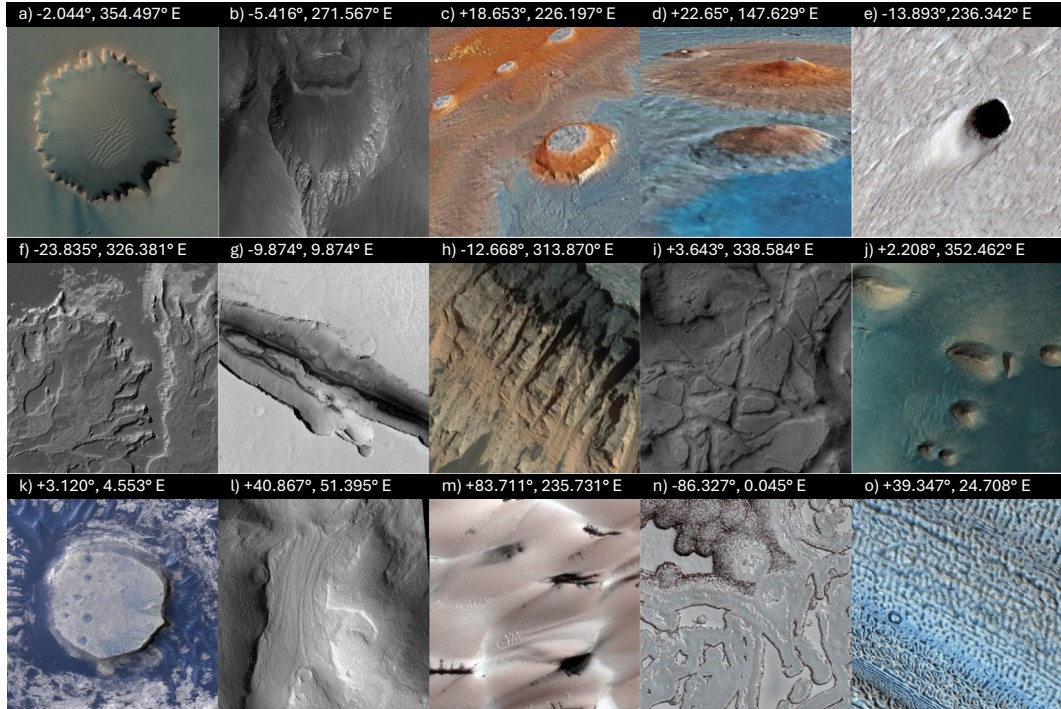

a) -2.044°, 354.497° E  b) -5.416°, 271.567° E  c) +18.653°, 226.197° E  d) +22.65°, 147.629° E  e) -13.893°,236.342° E
f) -23.835°, 326.381° E  g) -9.874°, 9.874° E  h) -12.668°, 313.870° E  i) +3.643°, 338.584° E  j) +2.208°, 352.462° E
k) +3.120°, 4.553° E  l) +40.867°, 51.395° E  m) +83.711°, 235.731° E  n) -86.327°, 0.045° E  o) +39.347°, 24.708° E

**Fig. 4 | Orders of relief and representative Martian terrain types relevant to aerobots.** (Top) A non-exhaustive hierarchical classification of the 2nd, 4th and 5th orders of relief of Mars, based on ref. 41. Some features listed in the classifications have picture examples shown in the bottom panels. (Bottom) Distinct topographic features and terrains of Mars with a potential for future science missions by aerial robots. Mars Reconnaissance Orbiter images are publicly accessible on the HiRISE web catalogue[47]. **a** Victoria Crater at Meridiani Planum with scalloped shaped rim and floored with dunes (TRA_000873_1780). **b** Non-circular 3 km impact crater on Tithonium Chasma's inclined wall (ESP_044998_1745). **c** False coloured 3D rendered view of Olympus Mons, Arsia Mons, Pavonis Mons & Ascræus Mons. Generated in JMars5 (GIS)[48]. **d** False coloured 3D rendered view of Elysium Rise showing Hecates Tholus (front), Elysium Mons & Albor Tholus. Generated in JMars5. **e** A pit crater about 50 m across formed over an underground lava tube (ESP_065887_1660). **f** Delta structure showing distributary network including lobes, inverted channels, & meander cutoffs in Eberswalde

Crater (PSP_001336_1560). **g** Cerberus Fossae East of the Head of Athabasca Valles (ESP_016216_1900). **h** Gully-like landslides, including alcoves and channels, on the interior wall of Valles Marineris (ESP_022632_1670). **i** Chaotic terrain of Aram Chaos showing rough floor topography with large, slumped blocks & fractures (PSP_008311_1835). **j** Mesas, buttes & yardangs surrounded by sand dunes in Meridiani Planum (ESP_034129_1820). **k** Inverted Crater and dunes in Arabia Terra (ESP_016459_1830). **l** Alpine glaciers, common at middle latitudes (ESP_019213_2210). **m** Dunes in the North covered primarily with ice, during springtime get topped with dark sand streaks due to geyser-like eruptions of carbon dioxide (ESP_017043_2640). **n** Swiss cheese terrain formed by circular & blob-shaped depressions at the South Pole due to the gradual ongoing sublimation of the CO2 ice layer (PSP_005095_0935). **o** Brain terrain is icy lobate features, in the image layering a small hill (ESP_033165_2195). Images Credit: NASA/JPL-Caltech/UArizona (image id number next to each image caption). NASA material/public domain.

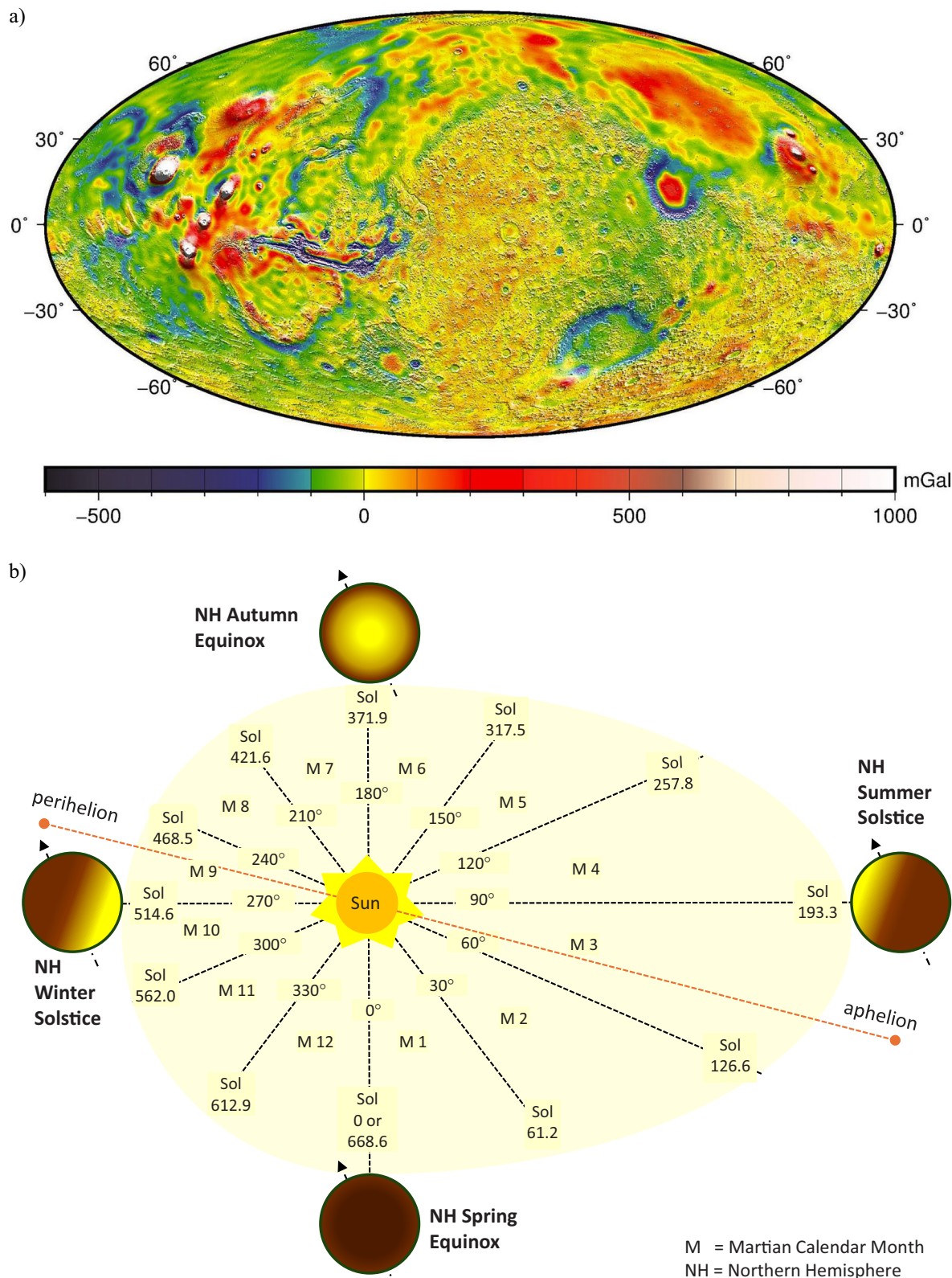

**Fig. 5 | Mars gravity-field variation and seasonal orbital geometry. a** Free-air gravity anomaly map of Mars from NASA's Goddard Mars Model-3 (GMM-3) (units: mGal; 1 mGal = $10^{-5}$ m s$^{-2}$). Image: NASA/GSFC[52] (public domain). **b** Schematic of Mars' orbit and seasonal geometry, showing perihelion/aphelion and key northern-hemisphere seasonal markers (not to scale).

poles, with $CO_2$ snow observed by Phoenix in 2008 sublimating before surface contact[57].

These variations require aerobots to adapt lift generation and thermal management to changing density and temperature regimes. Lower pressures in winter (especially in the south) reduce lift, demanding higher rotor speeds or altered flight profiles; extreme diurnal and seasonal temperature ranges emphasise robust thermal control. Seasonal mass exchanges at the poles also induce minor temporal gravity variations, further reinforcing mission planning that accounts for time, location, and elevation.

**Mars atmospheric structure.** Mars has a vertical structure broadly analogous to Earth's but lacks a stratosphere because ozone is absent[56]. Most weather occurs in the troposphere, which reaches 45 km (about three times higher than on Earth[41])—while the lowest, most dynamic layer is the planetary boundary layer, where atmosphere–surface exchanges dominate and local topography exerts strong control[58]. Understanding this structure is essential for predicting environmental conditions, ensuring EDL safety, and optimising aerobot design and operations[57]. Likewise, aerial robots can supply higher-resolution measurements of the lower atmosphere than orbiters or stationary surface assets, improving models of climate and surface–atmosphere interaction.

The following subsections within this discussion explore key atmospheric characteristics that influence aerobot performance, including boundary layer turbulence, wind patterns, and air density variations.

*Planetary boundary layer.* The planetary boundary layer (PBL) influences heat, momentum, and mass exchanges between the surface and atmosphere, directly affects flight conditions, sensor performance, and mission planning. Its influence on Martian weather and climate is amplified by the planet's thin, clear atmosphere, where surface temperature and drag variations have outsized effects due to low thermal and dynamical inertia[58]. Daytime conditions feature intense convection, with plumes and vortices reaching up to 10 km and following a dry adiabatic lapse rate of 4.3 °C per km—the temperature decrease of dry air with altitude[56]. The average lapse rate is lower (about 2.5 °C per km) because suspended dust absorbs solar radiation and warms the air. At night, convection weakens, radiative cooling forms a shallow stable PBL, and turbulence near the surface dominates[58]. PBL depth increases over elevated terrain due to stronger surface–atmosphere interactions and greater solar heating, producing more turbulence and complex wind patterns.

These dynamics have direct implications for Mid-Air Deployment (MAD) concepts proposed for next-generation Martian rotorcraft[59]. Unlike conventional landers requiring flat terrain, MAD can deploy few hundred meters above the surface from an aeroshell, reducing risks from thin-atmosphere aerobraking and high-impact landings. This enables access to highlands (>5 km above datum) where steep slopes and rough, unstable ground preclude conventional landing. By bypassing these constraints, MAD substantially expands accessible science targets for rotorcraft.

*Winds.* Wind shear and turbulence within the PBL challenge stable flight. Without oceans, Mars' circulation is simpler: equatorial regions have lower surface pressure from convective uplift; cold polar regions have higher pressure from subsidence[41]. A Hadley cell forms at low latitudes, driving surface winds from NE (north) and SE (south), while interactions at higher latitudes produce weather fronts and storms[57]. Typical seasonal wind velocities are 2–7 m s$^{-1}$ in summer to 5–10 m s$^{-1}$ in fall[51]. When wind exceed 17 m s$^{-1}$, it mobilises sand and fine particles, with storms reaching up to 30 m s$^{-1}$[60]. Despite high speeds, the thin air yields low dynamic pressure, so winds would feel like a light breeze to humans[61]. These regimes inform flight envelopes and operational constraints.

*Air density.* The primary parameter influencing most aerodynamic calculations for a Martian aircraft is the air density at its operational altitude. Mars' average surface air density, about 0.020 kg m$^{-3}$—less than 2% of Earth's sea-level density (1.225 kg m$^{-3}$)[51]—poses substantial challenges

for lift generation, necessitating larger rotors or wings. This value is optimistic, as actual density varies with location, altitude from datum, season, and time of day. Youhanna et al.[36] outlines two additional methods for more precise estimation. The first uses NASA's Mars atmospheric mathematical model[62], which calculates density by accounting for altitude effects with separate formulations for the lower (<7 km) and upper atmosphere. While conservative, the model does not incorporate regional or temporal variations. The second applies the Mars Climate Database (MCD v6.1)[63], a publicly accessible European-developed tool widely used in aerobot design studies[64,65]. It generates density estimates based on input parameters such as flight time and location and can output results as ranges or curves. Alternatively, the Mars Global Reference Atmospheric Model (Mars-GRAM)[66], primarily used by NASA and authorised partners, offers similar capabilities.

Figure 6 generated using MCD v6.1[63] for average solar conditions in Martian climatology, illustrates key atmospheric parameters—air density, viscosity, wind speed, and temperature—that directly influence lift, drag, stability, and power demands. Kinematic viscosity, derived from viscosity and density, determines the Reynolds number, which predicts laminar-to-turbulent transition and affects propulsion efficiency, lift, and stability. On Mars, Reynolds numbers are typically lower than terrestrial equivalents, so the flow can deviate from conventional Earth-aircraft assumptions and, in some cases, approach the regime of terrestrial insect flight—motivating adapted aerodynamic design approaches. Variations in density and viscosity alter aerodynamic efficiency; wind speed impacts control; and temperature fluctuations influence battery performance and material durability, making thermal management essential. Understanding these conditions is vital for optimising aerobot design for stable, efficient operation in diverse Martian environments.

**Martian dust effects.** Mars experiences dust devils (area covering <10 m²) and dust storms (~10⁶ m²) on a daily frequency lasting for short periods, driven by strong daytime turbulence that stirs surface dust[41,56]. Dust devils—often exceeding 12 km in height—are larger and more energetic than on Earth and occur commonly on warm, clear afternoons; they loft dust but contribute less to global events because increased atmospheric dust cools the surface and damps convection (negative feedback)[58]. However, dust storms create a complex environment that occasionally escalate into regional (~10⁶ km²) or global (>10⁶ km²), typically originating in the southern spring/summer, can persist for weeks to months, altering sunlight and temperatures and challenging solar-powered operations[56,58]. Overall, Mars' global atmospheric patterns are simpler and more predictable than Earth's[61], with NASA's monitoring revealing consistent seasonal dust activity[67]—an insight advantageous for aerial robot operations.

Besides natural events, rotorcraft operations also generate dust clouds. Ingenuity lifted about 0.1% of its mass in dust per flight, far higher than Earth helicopters (~0.02%), due to Martian environment[68]. Larger VTOL aerobots will likely raise proportionally more dust, posing risks of mechanical wear from abrasive particles and clogging or degradation of components. Martian dust, averaging 3 microns in size, can infiltrate non-sealed parts, necessitating robust protective designs[60]. This is critical for future crewed vehicles, as inhalation of dust containing silicates and oxychlorine compounds poses severe health risks[69]. Hence, Mars sample-return missions remain a top priority for NASA and other space agencies, as analysing these soils is critical for assessing the planet's viability for human exploration[37].

Electrification accompanies dust motion. Charge transfer during collisions and atmospheric separation can generate electric fields and occasional discharges, though limited by Mars' weak breakdown fields (~20 kV m$^{-1}$ vs ~3 MV m$^{-1}$ on Earth)[58,70]. Triboelectric charging around fast rotors may produce small fields or faint glows, as observed with Ingenuity[71]. Unlike Earth's moist ground, Mars' dry surface cannot dissipate charges, allowing accumulation and possible sparking with fast-moving vehicles[72,73]. Historical rover data showed wheel-charge buildup, mitigated with tungsten needles to safely discharge electricity[73]. Current evidence indicates Mars Helicopter faced

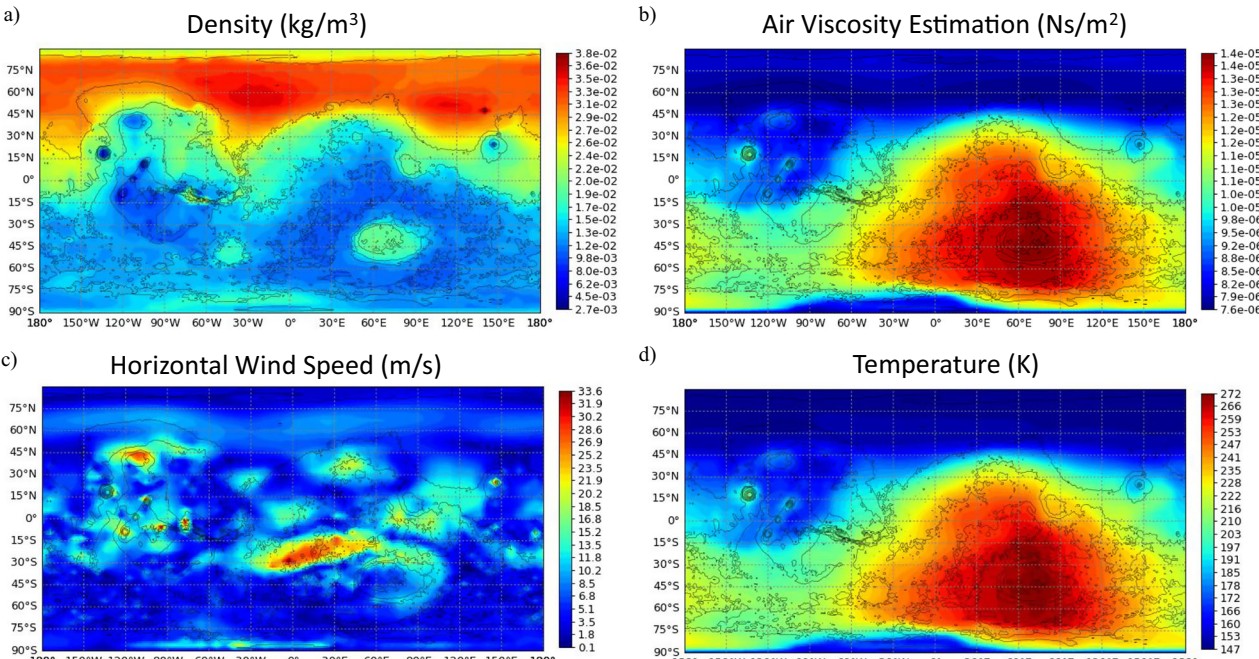

**Fig. 6 | Representative near-surface atmospheric conditions from the Mars Climate Database.** The graphs show global environmental parameters on Mars at the altitude of 10 m above the local surface of Mars at Longitude 0° Local time 11:00 on Sol 514.6 (Ls 270) during the Northern Hemisphere Winter Solstice. The panels show **a** air density, **b** air viscosity, **c** horizontal wind speed, and **d** temperature are parameters based on the climatology average solar scenario generated in MCD_v6.1[63].

minimal risk from static[71], larger high-speed aerobots will require shielding to protect electronics from potential circuit damage and signal errors[72].

Dust adhesion is another operational challenge, occurring mainly through electrostatic forces, with contributions from magnetic minerals like magnetite and weak van der Waals forces[72]. Dust on sensors and optics reduces visibility and GNC (Guidance, Navigation, and Control) accuracy, alters aerodynamics, adds weight, and accelerates mechanical wear, which impacts mission effectiveness and longevity. For solar-powered missions such as Insight, Opportunity, and Ingenuity; dust buildup reduces solar panel efficiency, power generation, and operational time, with Ingenuity even losing contact with Earth when dust coverage hindered battery charging[74].

**Martian radiation effects.**

*A detailed discussion of Mars' radiation environment, electronic failure modes, shielding materials, and human health limits is provided in the* Supplementary Information – Notes *(SN1).* (*refs.* [75–87]).

Mars is subject to significant particle and electromagnetic radiation due to its lack of a global magnetic field and ozone layer[14]. Particle radiation includes high-energy galactic cosmic rays (GCRs) originating from outside the solar system and lower-energy Solar Energetic Particles (SEPs) emitted during solar storms[72,75]. The solar cycle modulates this exposure: solar maximum increases SEP events while reducing GCR influx, whereas solar minimum allows higher GCR penetration[76]. These radiation types can degrade spacecraft materials, damage electronics, and increase risks for biological payloads and human crews[72,75]. For electronics, effects include total ionizing dose (TID) degradation, displacement damage in semiconductors, and Single Event Effects (SEE) ranging from transient glitches to permanent failures[76]. Electromagnetic radiation, particularly ultraviolet (UV) light, poses additional long-term degradation risks to materials[75,77].

Shielding strategies traditionally rely on aluminium, but advanced hydrogen-rich composites and boron nitride nanotubes show promise for reducing both mass and secondary particle generation[79]. Radiation Hardening Assurance combines shielding, hardened components, redundancy,

and rigorous testing[80]. Ingenuity's avionics illustrate one mitigation approach: a layered architecture combining a radiation-tolerant field programmable gate array (FPGA) for low-level tasks, dual-redundant automotive processors for control, and a high-performance Snapdragon 801 processor for image processing, compensating for its lack of hardening with parallel redundancy[29].

Beyond uncrewed aerial systems, there have been conceptual studies, such as in refs. 84 and 85, where radiation exposure critically influences shielding requirements, mission duration, and operational altitude constraints. For human missions, exposure limits are stringent, with average surface dose rates on Mars around 90–100 times higher than Earth's background[85]. Lower elevation landing sites offer additional atmospheric shielding[87], an important consideration for both human exploration and aerobot system longevity.

**Martian solar irradiance.**

*A detailed account of irradiance modelling, PV design parameters, degradation mechanisms, and dust mitigation technologies is given in the* Supplementary Information – Notes *(SN2).* (*refs.* [88–95]).

Mars receives about 590 W m$^{-2}$ of solar energy at the top of its atmosphere—roughly 57% of Earth's—due to its greater solar distance[51,88]. Atmospheric dust further reduces and scatters incoming sunlight, with optical depth ($\tau$) ranging from 0.5 in clear skies to above 3.0 in severe dust storms. Direct sunlight is optimal for power generation, but during dust events diffuse irradiance dominates, and global irradiance can fall from ~400 W m$^{-2}$ on a clear summer day to ~80 W m$^{-2}$ in a winter storm[88].

Photovoltaic (PV) systems are well-suited for aerobots due to their low mass compared with nuclear systems. Fixed-tilt arrays are simple but less efficient than tracking systems, and flat-plate PVs capture both direct and diffuse light, maintaining power during dusty periods. High-efficiency III–V multi-junction solar cells, such as Ingenuity's quadruple-junction array[29], outperform silicon cells in both efficiency and radiation tolerance, benefiting further from Mars' low ambient temperatures[88,90,93]. However, dust deposition remains the main degradation factor, reducing output by ~0.28%

per sol, with long-term missions showing a mix of reversible (wind removal) and irreversible losses[88,93,94]. Mitigation strategies such as redundancy, real-time power management, optimised tilt angles, and durable coatings are essential for sustaining PV performance over multi-year missions.

## Design, development and delivery considerations for Martian aerobots

Comprehending Mars' atmospheric and environmental conditions is crucial for designing in situ aerial robotic missions. These insights guide the development of exploration technologies adapted to the planet's unique challenges. The thin atmosphere demands highly efficient lift mechanisms and strong, lightweight structures for low-pressure flight. Autonomous capability is critical due to communication delays with Earth, requiring energy-efficient, radiation-hardened systems capable of independent navigation without Global Navigation Satellite System (GNSS)[96]. Limited digital maps, frequent dust storms, and variable wind and pressure patterns make onboard sensing, real-time terrain mapping, and adaptive navigation indispensable. Seasonal planning further optimises performance by targeting periods of higher atmospheric pressure and environmental stability. The Ingenuity helicopter operated semi-autonomously on Mars, self-managing navigation, stability, and landing, while relying on human input for mission planning and flight parameter definition[29]. Aerobots must use ultra-light, durable materials to withstand extreme temperatures ($-153\,°C$ to $20\,°C$)[12], resist dust-induced corrosion and triboelectric interference, and incorporate reliable energy and thermal management to counter solar efficiency loss and winter cold. Topographic variability in relief and surface roughness further influence stability and energy consumption, shaping sortie envelopes and landing/take-off site selection. Overall, success depends on coordinated advances across aerodynamics, propulsion, materials, autonomy, and space-system engineering to deliver reliable, repeatable Mars operations.

### Ingenuity into design and development considerations

Integrated design methodologies that align mission requirements, vehicle design, and environmental factors are critical for developing Martian aerobots. This process uses specialised design and simulation tools, supported by rigorous testing in simulated Mars environments and Earth-based analogues. The first Mars Helicopter, Ingenuity, faced substantial challenges, demanding a precise balance of structural integrity, thermal management, aerodynamic efficiency, and contamination control to protect sensitive components and meet planetary protection standards[97]. Engineering innovations—such as rotor optimisation, lightweight landing gear, and ruggedised electronics for extreme conditions—enabled effective operation in Mars' environment. Comprehensive design, testing, and validation, including flight, structural, and environmental assessments, confirmed Ingenuity's resilience to space travel, Martian conditions, and operational stresses, securing its role as a technology demonstrator for future missions. The following subsections outline these challenges and the key design, development, and testing considerations for successful aerobot deployment.

**Flight dynamics considerations**. Mars' atmosphere, at about 1% of Earth's density, poses significant challenges for aerial vehicle design. Terrestrial combustion engines are impractical due to the lack of free oxygen, although the thin atmosphere still allows for rotary propulsion systems like rotors or propellers[96]. The thin air significantly reduces the Reynolds number, which measures the ratio between inertial and viscous forces in the atmosphere, leading to decreased aerodynamic efficiency. A lower Reynolds number increases the possibility of flow separation and drag over the flight surfaces, making it more difficult to generate lift and maintain stable flight in Mars' atmosphere. Additionally, the speed of sound on Mars is 70% lower than on Earth[38], meaning that rotary aerodynamic surfaces are more prone to approaching supersonic speeds, potentially causing compressibility effects like shock waves that further degrade performance. While existing literature, including data from Ingenuity's flights, offers valuable insights into these phenomena,

empirical data remains limited. This limitation complicates the design process, necessitating extensive research and specialised airfoil designs optimised for Mars' unique aerodynamic conditions[96]. Another significant challenge is the reduced aerodynamic damping in Mars' thin atmosphere, which affects the rotor blades' ability to control blade flapping, the up-and-down motion of the blades caused by cyclic aerodynamic and inertial forces during flight. On Earth, the denser atmosphere provides sufficient aerodynamic resistance to manage this flapping motion, but on Mars, this damping is reduced by about ten times, leading to oscillatory flap modes that could destabilise the rotary aerobot by coupling with its body during flight[97].

To address Mars' thin atmosphere, the Mars Helicopter's rotor blades were designed using advanced rotor aerodynamics tools to optimise performance under low Reynolds numbers. The initial geometries were refined for structural integrity and efficiency, with specialised software predicting aerodynamic performance. Thin laminar flow airfoils were utilised to minimise drag while ensuring structural strength, which is critical for stable flight in the Martian environment. To mitigate the reduced aerodynamic damping, the rotor blades were made extremely stiff, raising the natural flap mode frequencies to a higher range, where they are less likely to interfere with the helicopter's control system[97].

*Flight testing*. The flight testing of the Mars Helicopter, Ingenuity, was conducted in the NASA's Jet Propulsion Laboratory (JPL) Space Simulator, a cylindrical chamber 7.6 m in diameter and 24.4 m in height, designed to replicate Mars' low atmospheric pressure and gravity[97]. The chamber was evacuated to a near-vacuum and filled with $CO_2$ to simulate the Martian atmosphere and equipped with precision instruments including a multi-directional force/torque sensor and motion tracking systems. Tests assessed lift generation, hover stability, control responsiveness, and external forces in simulated Martian conditions, including low-speed flight and aerodynamic stability under controlled wind speeds of up to $10\,m\,s^{-1}$ [97]. Unique to Mars rotorcraft, the testing involved a gravity offload system designed to simulate Mars' gravity, about 38% of Earth's ($3.71\,m\,s^{-2}$ compared to $9.81\,m\,s^{-2}$). This system, using pulleys and counterweights, reduced the helicopter's effective weight, allowing it to perform free-flight tests within the chamber[97]. This setup enabled Ingenuity to simulate flight conditions on Mars, including take-off, hovering and manoeuvring in a controlled environment. Additional testing related to spacecraft environmental conditions, such as vibration and shock from launch and landing, was performed in a smaller 3-m chamber.

Some other specialised facilities extend Martian flight research. Tohoku University's wind tunnel replicates low Reynolds numbers and high subsonic flows, enabling studies of flow separation and compressibility effects critical for rotorcraft optimisation[98]. The California Institute of Technology's low-density multi-fan wind tunnel investigates forward-flight dynamics and wind interaction[99]. At the Harbin Institute of Technology, a reflux subsonic low-density dust wind tunnel simulates Martian dust storms, allowing analysis of erosive and abrasive effects on rotorcraft durability[100].

**Structural material and design considerations**. The Martian environment poses severe thermal challenges and demands strict planetary protection to prevent contamination with Earth-based microorganisms[96]. Aerobots operating in temperatures that can drop below $-100\,°C$ require materials resistant to brittleness, excessive contraction, and microcracking[97]. Minimising outgassing—the release of trapped gases in low-pressure environments—is essential to prevent contamination. Carbon fibre composites, valued for their high strength-to-weight ratio, were used in Ingenuity's rotor blades and landing gear, with a specially formulated resin to reduce outgassing. The landing gear, designed for uneven terrain, featured long deployable composite legs providing a broad base to minimise tipping, with integrated flexures and dampers to absorb impacts, and a mechanism ensuring reliable extension and locking. The solar array, mounted above the rotor blades (Fig. 7), recharged lithium-ion batteries and shared structural features with the

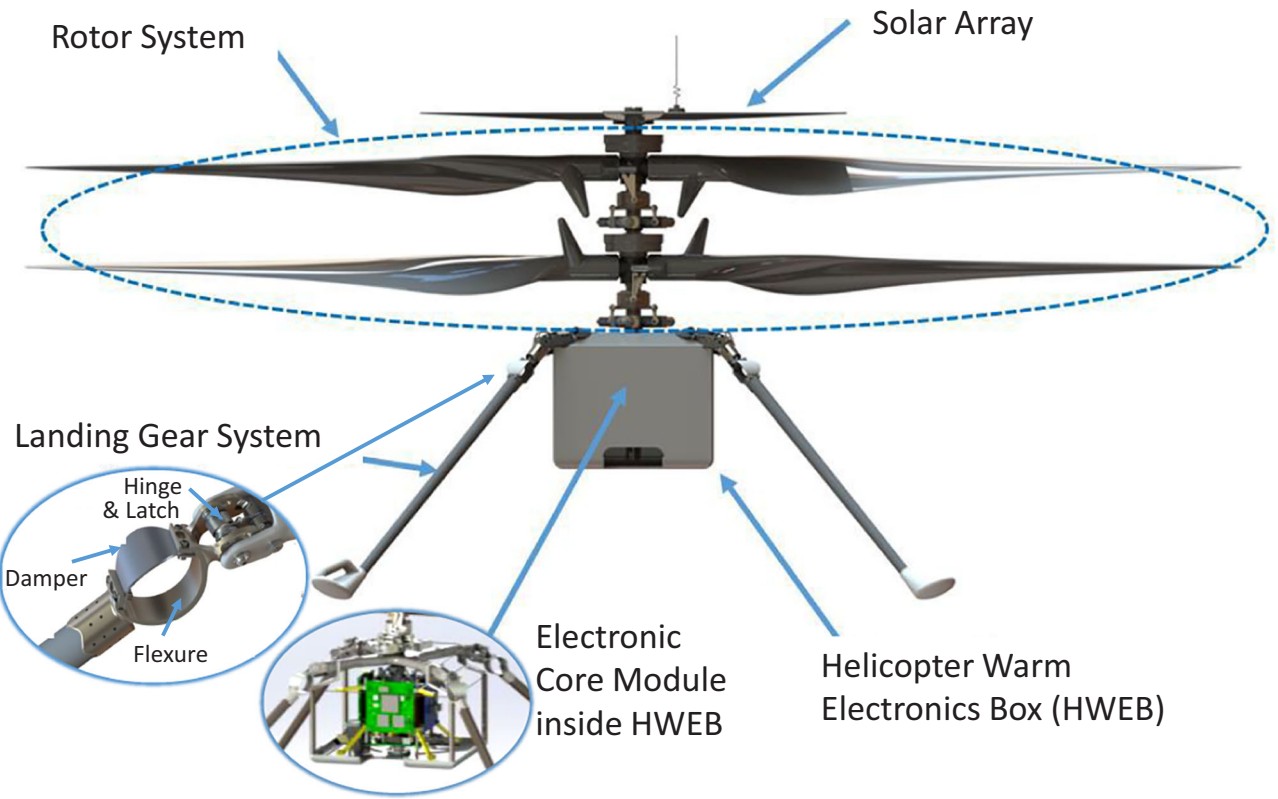

**Fig. 7 | Mars Helicopter technology demonstrator subsystem classification.** Adapted from reference source[97], by permission of the American Institute of Aeronautics and Astronautics, Inc.

blades—moulded carbon fibre composite skins, foam core, and a central hub interfacing with the rotor system.

Thermal management is critical on Mars due to its low atmospheric density, which limits heat dissipation[96]. Ingenuity's beryllium–aluminium motor hub acted as a heat sink, managing propulsion and electronics heat while stabilising thermal expansion[97]. The Helicopter Warm Electronics Box (HWEB) insulated electronics and batteries, prevented contamination, and met planetary protection requirements. Constructed from a high-absorptivity, low-emissivity polyamide film around a lightweight composite frame, it created an insulating gas gap to reduce heat loss, while optical windows provided clear views for the cameras and laser altimeter.

*Structural testing.* Structural testing is essential for verifying the mechanical strength and durability of aerobot components under mission-specific loads and stresses. Finite Element Analysis (FEA) is commonly used to simulate and predict structural behaviour under conditions such as launch forces, centrifugal forces on rotor blades, and thermally induced stresses from material expansion differences. For the Mars Helicopter, these simulations were validated through experimental tests to ensure the design could endure the interplanetary journey, deployment on Mars and operational stresses of the Martian environment[97].

The integrity of the rotor system is confirmed through destructive load testing, which evaluates the structure's ability to withstand extreme forces, and modal testing, which identifies the natural frequencies and vibration modes. In the case of Ingenuity, rotor blades were tested to ensure they could withstand the high centrifugal forces experienced during flight[97]. Landing gear, designed for stability on uneven terrain, undergoes drop tests with scale models to emulate various landing scenarios. For the Mars Helicopter, these tests, using the similar gravity offload system used in the flight testing, included evaluations on surfaces like concrete, sand, and rocks, as well as sloped terrain, ensuring the gear could absorb impacts without tipping. Its solar array was tested at JPL's Environmental Test Laboratory to verify its structural resilience through both modal and random vibration tests, ensuring it could withstand launch and operational vibrations.

Thermal vacuum testing is critical for ensuring an aerobot can operate in extreme Martian conditions. The Mars Helicopter was tested in a chamber that simulated Mars' environment by creating a near-vacuum and backfilling it with carbon dioxide to simulate the thin atmosphere[97]. The temperature within the chamber was cycled to simulate the daily temperature variations on Mars, with diurnal changes exceeding 100 °C[56]. This testing was vital for verifying the function of HWEB for insulating the sensitive electronics packaged inside it.

## Interplanetary delivery considerations

In addition to atmospheric and operational constraints on Mars, practical mission design must ensure that aerobots can be safely delivered from Earth to the Martian surface within existing launch, EDL and packaging architectures. This subsection outlines the key delivery constraints imposed by launch and cruise loads, aeroshell and lander geometry, future transport capabilities and overall launch mass budgets.

*Interplanetary travel stresses.* Aerobots designed for planetary exploration must withstand extreme forces during launch and throughout their journey to the target planet. During the launch phase, spacecraft can experience steady-state accelerations of 5 to 7 g, while during entry, descent, and landing (EDL) on Mars, g-loads can peak between 9 and 16 g, depending on the launch vehicle, its mass and specific mission configurations[101]. The Mass Acceleration Curve is a critical guideline that determines the maximum acceleration forces components must endure based on their mass, ensuring robustness against these intense conditions.

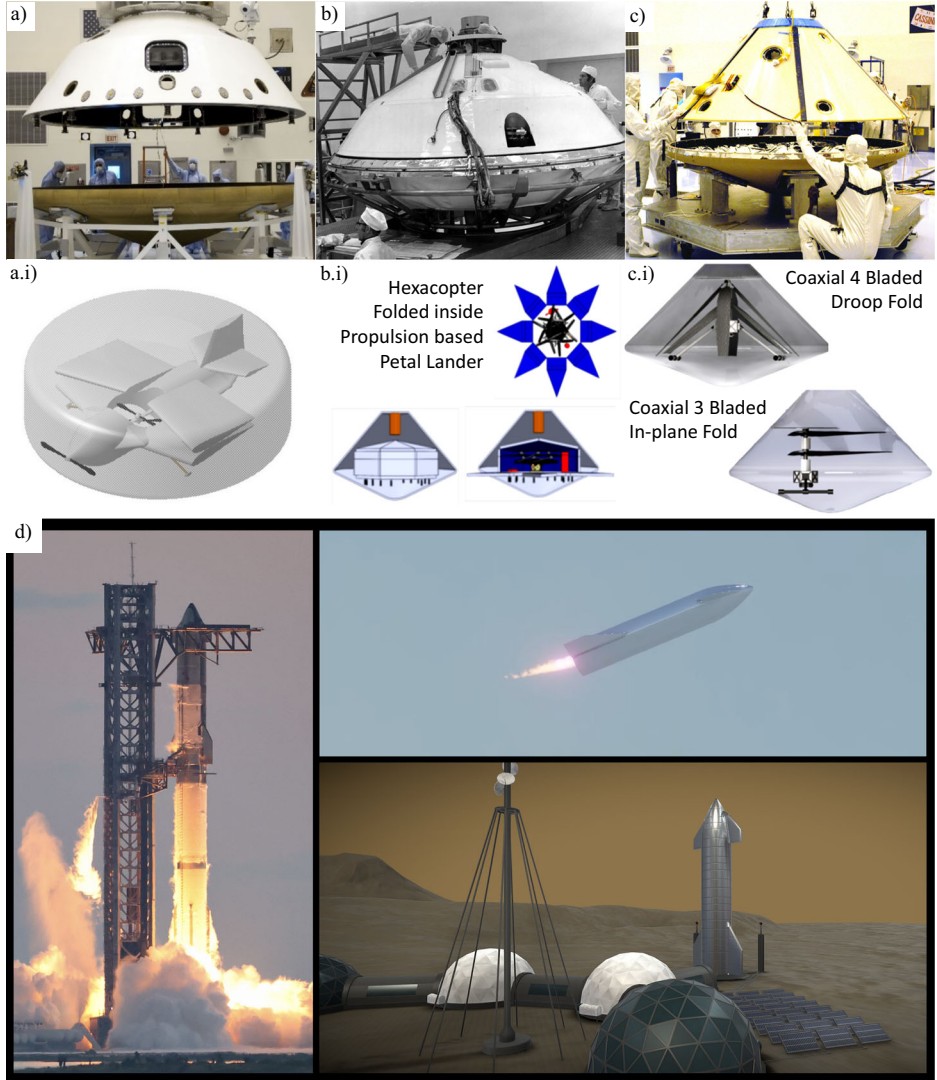

**Fig. 8 | Aeroshell packaging constraints for Mars aerial vehicles and example concepts.** The top images depict Mars's past mission aeroshells during the testing phase: **a** Mars Science Laboratory aeroshell (2011, size 4.5 m). **b** Mars Viking 1 aeroshell (1975, size 3.5 m). **c** Mars Exploration Rover 2 aeroshell (2003, size 2.65 m). Images Credit: NASA (public domain). The images in the middle row are CAD model illustrations showing different aircraft folded and packed inside the respective aeroshell shown at the top: **a**.i Winged VTOL concept CAD model 4.5 m aeroshell[106]. **b**.i Folded hexacopter concepts folded inside a petal lander comprising thrusters which are packed inside the Viking aeroshell (3.5 m). **c**.i Coaxial rotorcraft with 3 and 4 blades variants packed inside 2.65 aeroshells. Images credit (**b**.i–**c**.i): NASA material/public domain, reproduced/adapted from[104] (NTRS 20200002139). **d** SpaceX Starship illustrations: (left) launch-site during test flight; (top right) in-flight "belly-flop" render (screenshot); (bottom right) Mars base concept render. Credits: Steve Jurvetson (Flickr, CC BY 2.0); janos (YouTube, CC BY 3.0); MOJackal (Sketchfab, CC BY 4.0).

The Mars Helicopter was designed to endure up to 60 G quasistatic acceleration[97], a limit providing a high safety margin which accounts for extreme scenarios such as hard landings, deployment shocks, and unexpected impacts. In addition, the root mean square of acceleration (Grms), measures the intensity of vibrations and addresses the continuous vibrational stresses experienced during launch. Ingenuity was sized for 7.9 Grms to help prevent components from entering resonance, which could lead to structural failure[97]. These factors necessitate significantly more structural mass for an aerobot than would be required for flight operations alone.

Ingenuity underwent rigorous testing at the JPL to ensure its structural integrity, including vibration tests to simulate launch forces, modal tests to avoid resonance, and shock tests to replicate impacts during separation and landing. It was mounted on a vibration table and subjected to random vibration levels simulating launch conditions, including attachment to its delivery system beneath the Perseverance rover[97]. Post-test inspections confirmed the helicopter's durability. Shock testing simulated the forces during separation from the launch vehicle and deployment from the rover,

using controlled devices to replicate the detachment process. These comprehensive tests validated the components of the rotorcraft could withstand these mechanical shocks, ensuring its readiness for the most challenging phases of the mission.

**Spacecraft transport package sizes.** The size and design of aircraft for Mars missions are critically constrained by the EDL system. As the spacecraft enters Mars' atmosphere with a descending speed of 6 km s⁻¹, the protective covering called aeroshell shields it from extreme heat and shock waves while also imposing strict geometric limitations on the aerobot's maximum size[102]. Historically, NASA has utilised three different aeroshell sizes for successful EDL systems, as illustrated in Fig. 8 (top row). The smallest aeroshell, 2.65 m in diameter, is used in several key missions, including the Mars Exploration Rovers (2003) and InSight (2018). The next size up is 3.5 m, which was used exclusively for the Viking 1 and 2 Landers (1975)[103]. The largest aeroshell, first used in the Mars Science Laboratory (2011) and later in the Mars 2020 mission[103], sets the current

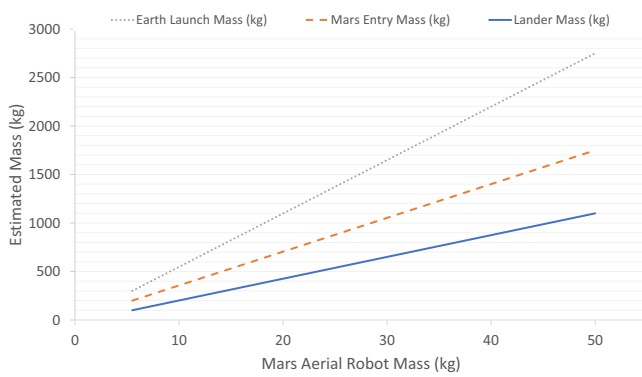

**Fig. 9 | Mass-scaling relationship between Mars aerial robot mass and estimated Earth launch, Mars entry, and lander mass.** Plot redrawn based on Young (2000, Fig. 10)[97].

size limits for Mars aerobots, constraining them to a maximum diameter under 4.5 m and a height of approximately 2.2 m[104].

The unique challenge for aerial vehicles due to the thin atmosphere on Mars, necessitates larger wings or rotor blades for heavier-than-air craft and oversized balloons for lighter-than-air craft to achieve sufficient lift. For instance, NASA's Ingenuity helicopter, which weighs only 1.8 kg, has long rotor blades measuring 1.2 m in diameter[29]. While Ingenuity's compact design was adequate for its limited mission as a rover appendage, carrying a more substantial payload would require a significantly larger aircraft, which in turn demands more space for larger wings or blades. To accommodate these larger flying surfaces within the constraints of the aeroshell, it becomes essential to incorporate folding mechanisms into the aerobot's design, allowing efficient stowage and deployment upon arrival. As shown in Fig. 8 (middle row), different aerobot configurations, including winged VTOL[105] and coaxial rotorcraft designs, demonstrate how folding structures enable compact packing while maintaining structural integrity and operational efficiency. These mechanisms allow wings or blades to remain stowed during transit and deploy once the aerobot reaches Mars. Since the constraint is on the physical size of an aerobot rather than its weight, which must remain light to fly on Mars, there is sufficient space to accommodate a smaller lander or rover alongside the aerobot. The petal lander, utilised in earlier missions like Pathfinder and MER rovers, can provide additional support to aerobots by absorbing loads during the liftoff and landing (Fig. 8b.i)[103]. The selection of landing systems, whether powered deceleration thrusters, airbag systems, or sky cranes, depends on specific mission requirements, such as payload size, terrain, and precision needed during the final descent and landing phases of soft EDL[102]. Although developing a new aeroshell and lander would be optimal, space missions often favour existing designs for their proven reliability[103].

**Futuristic transportation.** Advances in Martian aerobot capabilities are closely tied to improvements in interplanetary transportation. Larger systems, such as expanded aeroshells, could enable the delivery of heavier and more capable aerial platforms, extending mission durations and supporting in situ resource utilisation. Current aerobots, like Ingenuity, remain limited by existing Mars aeroshell size and mass constraints, but future large-scale transport could accommodate high-endurance platforms with advanced instruments, enhanced power systems, and greater mobility; substantially expanding scientific reach.

SpaceX's Starship spacecraft and Super Heavy rocket (Fig. 8d) exemplify such potential, offering a fully reusable system for crewed and cargo missions. The cargo variant is designed to deliver up to 150 tonnes in reusable mode or 250 tonnes expendable, with an internal payload space 8 m in diameter and 17 m in height[106,107]. Development focuses on rapid iteration[108,109], in-orbit refuelling, and Martian in situ resource use to enable sustainable round trips[110], paving the way for aerial vehicles far beyond current capabilities. However, given the complexity of space systems, such advancements may take considerable time to materialise.

**Table 1 | Summarised comparison between aerobot and rover**

| Attributes | Aerobot | Rover |
|---|---|---|
| Traverse Speed | High | Very Low |
| Obstacle Overpassing | High | Very Low |
| Dynamic Agility | High | Very Low |
| Specific Range | High | Very Low |
| Operational Endurance | Low | Medium |
| Payload Capacity | Low | High |
| Shape & Size Flexibility | Low | High |
| Robotic Capability | Low | High |

**Launch budget.** The optimal window that space agencies always use for launching a spacecraft to Mars, typically a 6–9-month journey, occurs every 26 months when Earth and Mars align near Mars' closest approach to Earth, sometimes coinciding with its perihelion (Fig. 5b)[104]. For Mars robotic missions, Young et al.[96] estimated that adding 1 kg to an air vehicle increases the lander mass by 21.5 kg, the entry mass by 13 kg, and the launch mass by 20 kg (as illustrated in graph Fig. 9). These figures, derived from the 1998 Mars Pathfinder mission, show that the 16 kg rover and its equipment led to a total launch mass of 890 kg.

Mission costs were estimated with a fixed launch vehicle cost of $70 million and an incremental mission development cost of $180,000 per kg of launch mass (in 1997 USD), resulting in a total cost of about $230 million for Mars Pathfinder[96]. Adjusted for inflation, $230 million in 1997 is equivalent to about $464 million today (2025), reflecting a cumulative price increase of ~102% over 28 years due to an average annual inflation rate of 2.54%[111]. While the inflation-adjusted figure provides a base estimate, the increased complexity and advanced technology required for modern Mars missions could raise costs by an additional 20–30%, potentially bringing the budget to $557 to $603 million. This estimate method aligns closely with the cost of the Phoenix mission (2007), which totalled $407 million, including $321 million for spacecraft development and $86 million for its launch, using the same Delta II rocket and a 2.65-m aeroshell[112]. Launch service costs can increase by 2.5 times for missions using larger rockets such as the Atlas V, as seen with the Mars Science Laboratory (MSL) and Mars 2020 projects, where development costs exceeded $2 billion due to large payloads and new methodologies[113]. Ultimately, the escalating costs and narrow launch window available only about every two years, highlight the importance of efficient mass management and strategic planning in future Martian aerobot missions. Limited payload capacity and high transportation costs demand that Martian aerobots be designed for maximum functionality with minimal weight, optimising energy efficiency, material selection, and payload integration to ensure mission feasibility within these constraints.

## Mars Aerobot Design Thinking Matrix

The Mars Aerobot Design Thinking Matrix provides a structured framework to align environmental constraints with mission objectives, ensuring coherent trade-offs across interdependent factors. It translates Mars-specific conditions into design choices that balance energy efficiency, flight stability, environmental adaptability, and operational feasibility. The matrix is intended to guide systematic decision-making from early concept to detailed design, maintaining focus on mission goals while constraining complexity. It also supports comparative assessment among candidate configurations and clarifies where margins are required. By consolidating constraints and objectives into an integrated scheme, the matrix helps engineers prioritise what matters for reliable operation on Mars and keeps designs closely focused on demonstrable mission value.

### The rationale for aerobots inclusion

The development of a Mars aerobot design involves an analytical thought process that is primarily based on the mission requirement to customise an optimised overall design by prudently examining the trade-offs amongst the

numerous parameters, design configurations and technologies. A reasonable first step before such a process would be a quick justification of the inclusion of aerobots in the Mars missions against current rovers. To remove any ambiguity, an aerobot here is defined as any kind of autonomous robot with lifting capability into the air. Table 1 shows the comparative analysis between aerobot and rover, by allocating a qualitative rating on a 5-point scale of very low to very high, against eight selected attributes. The ratings assigned are according to Martian vehicle standards, and some parameters could be rated in the form of a range, such as traversing speed can be ranged from high to very high, but the single most likely rating is mentioned for simplicity.

The lateral velocity of the small Ingenuity Helicopter is capped at $2 \, m \, s^{-1}$, with a capacity of going up to $20 \, m \, s^{-1}$, whereas its companion Perseverance rover can travel across a hard flat surface with a speed of $0.04 \, m \, s^{-1}$ [29,30]. This great disparity between travelling speed combined with the ability to travel upwards further gives aerobots a high edge against the rover in obstacle overpassing, dynamic agility, and specific range. Dynamic agility, in the context of Mars exploration, refers to an aerobot's ability to perform rapid, precise manoeuvres to adjust flight paths, avoid obstacles, and adapt to wind disturbances. Mars' thin atmosphere provides less aerodynamic damping, making stability and control more challenging than on Earth. Unlike rovers, which are confined to ground traversal, aerobots leverage their agility to explore diverse terrains efficiently, access high-priority scientific sites, and cover greater distances in shorter timeframes.

Rovers are heavy which limits their speed and movement but have greater flexibility in shape and size that benefits in accommodating more power, science payloads, and experimenting capabilities; for instance, the Perseverance rover is about 1025 kg, but it is a walking lab of volume size of $3 \times 2.7 \times 2.2 \, m$, carrying 7 science instruments, a long robotic hand, and a samples storage system [104]. Flying a rotary-propelled vehicle in the Martian thin atmosphere consumes significant power and produces in-system heat, due to the requirement of generating sufficient thrust aerodynamically—particularly for helicopter-type aerobots, whose rotor blades must operate with high tip speeds close to the local speed of sound. Consequently, Ingenuity has flying endurance of only over a couple of minutes [29]. Rovers on the other hand are slow but have higher operational endurance regarding continuous travel and experimentation. Given that rovers can last over a decade, they can overall cover a wider land when compared to a short-term aerobot mission. For missions that would include a combination of both land and aerial robots, the individual attributes of each machine need to be counted and coordinated in such a way that they justify and enhance the overall mission performance and output.

### Orders of Martian aerobot preliminary design

The formation of the Mars Aerobot Design Thinking Matrix is majorly based on the trends observed in the research literature [114] on Martian aerobot designs. These studies highlight key constraints such as aerodynamic performance in Mars' thin atmosphere, energy efficiency, take-off mechanisms, mission longevity, and mission types based on energy resources and operational objectives. Incorporating these insights, the matrix establishes a hierarchical selection process to define prerequisites for a preliminary aerobot design, as shown in Fig. 10 and discussed in the following subsections.

**Mission types.** In this order of selection (Fig. 10a), the foremost choice to make is between a disposable and reusable Mars mission-type aerobot design. The reusable mission type requires an aerobot that would be capable of VTOL and uses long-lasting or reusable fuel, as there are no runways and refuelling infrastructure on Mars; whereas the mission is deemed disposable when the aerobot lacks such reusable capability. However, a disposable mission would essentially require a scientific justification for its value in terms of planetary exploratory budget and resources. Each mission type influences the next choices in the series, accordingly.

After selecting the mission type, the next key decision is whether the aerobot will be rover/lander-dependent or fully independent, as this influences its design, launch mechanism, and power system. A rover/lander-dependent aerobot offers advantages like assisted launches, on-

ground instrumentation, and reduced weight due to shared resources. However, the slower pace of traditional rovers can limit the aerobot's efficiency and range. Recent advancements, such as NASA and Boston Dynamics' Mars Dog robot [115], could resolve this issue by matching the aerobot's speed, allowing for more efficient, synchronized operations. Nevertheless, this approach requires adapting the rover/lander to accommodate the aerobot. In contrast, an independent aerobot, while lacking support from a rover/lander, offers greater range, autonomy, and potentially faster mission completion because it is not constrained by the need to return to a fixed ground base for recharging, data transfer, or operational coordination. This allows it to cover larger areas in a shorter time, making it particularly advantageous for reconnaissance and large-scale mapping missions. The decision between dependency and independence should consider the specific mission objectives, balancing the benefits of support against the flexibility of autonomy.

Next in the order series is the aerobot system design that includes a set of three primary sub-systems at this level, each allocated with the known possible options. These subsystems are arranged in one of the feasible rational orders, comprising the launch mechanism, power source type, and aerobot type.

**Take-off mechanism.** The take-off mechanism, or system, for flight is a good starting point in the preliminary design phase as it helps to think of the best way of take-off as per mission requirements and hence, the required technology. There are 6 ways of launching an aerobot, as mentioned in Fig. 10a. and choosing one of them depends on the previous selections made. For instance, VTOL can only be for a reusable mission and VTOL from the station is only possible if the aerobot is developed to be rover/lander-dependent. VTOL from the ground is a versatile option that can be effective for both rover/lander-dependent aerobots, such as the NASA Mars Electric Reusable Flyer concept shown in Fig. 11a [116,117], and independent aerobots, as demonstrated by the Ingenuity Helicopter, which successfully took off from the ground after being deployed from the belly of the Perseverance Rover [104].

A catapult launch from a station, suitable for both disposable and reusable aerobots, involves propelling the aerobot into the sky. This method requires a rover or lander on the ground to support the launch mechanism, similar to those used in terrestrial tactical aerial drone systems [118]. Recovery of the drone can be a challenge in such take-offs, as drones without VTOL will be disposed of unless they make a return back to the station within the same flight. Reuseable drones, on the other hand, can utilise this strategy to increase range and altitude for specific missions. Similarly, aerobots launching from the station carrying rocket or propellant-based propulsion, are designed to be one-time missions. A terrestrial canister launched small OUTRIDER drone by Lockheed Martin is an example of such an unmanned aerial system [119].

Both mid-air deployment and rocket propulsion from the ground take-off mechanisms can be applied to any combination of preliminary selections within the Mars Aerobot Design Thinking Matrix. Mid-air deployment, especially for disposable aerobots, is one of the earliest concepts in planetary exploration, traditionally involving an independent, rocket-fuelled aerobot [120,121]. Recent research has focused on using mid-air deployment of VTOL aerobots to explore Mars' highlands before landing and re-flying [59]. Alternatively, rocket propulsion from the ground offers a more controlled and time-specific launch option. A reusable VTOL drone with an auxiliary rocket-fuelled propulsion system for high-altitude flight could also benefit from this ground-based launch mechanism when needed.

**Power source types.** The next step in the aerobot system design process (Fig. 10a) is selecting the appropriate power source. In the earliest Mars aerobot design concepts, the Hydrazine Engine has been a popular choice for a disposable mission-type aircraft due to its ability to provide high energy and propulsion thrust at a temperature of 926 °C with a minimal engine design complexity and monopropellant property, which is of advantage in an oxygen-deprived atmosphere

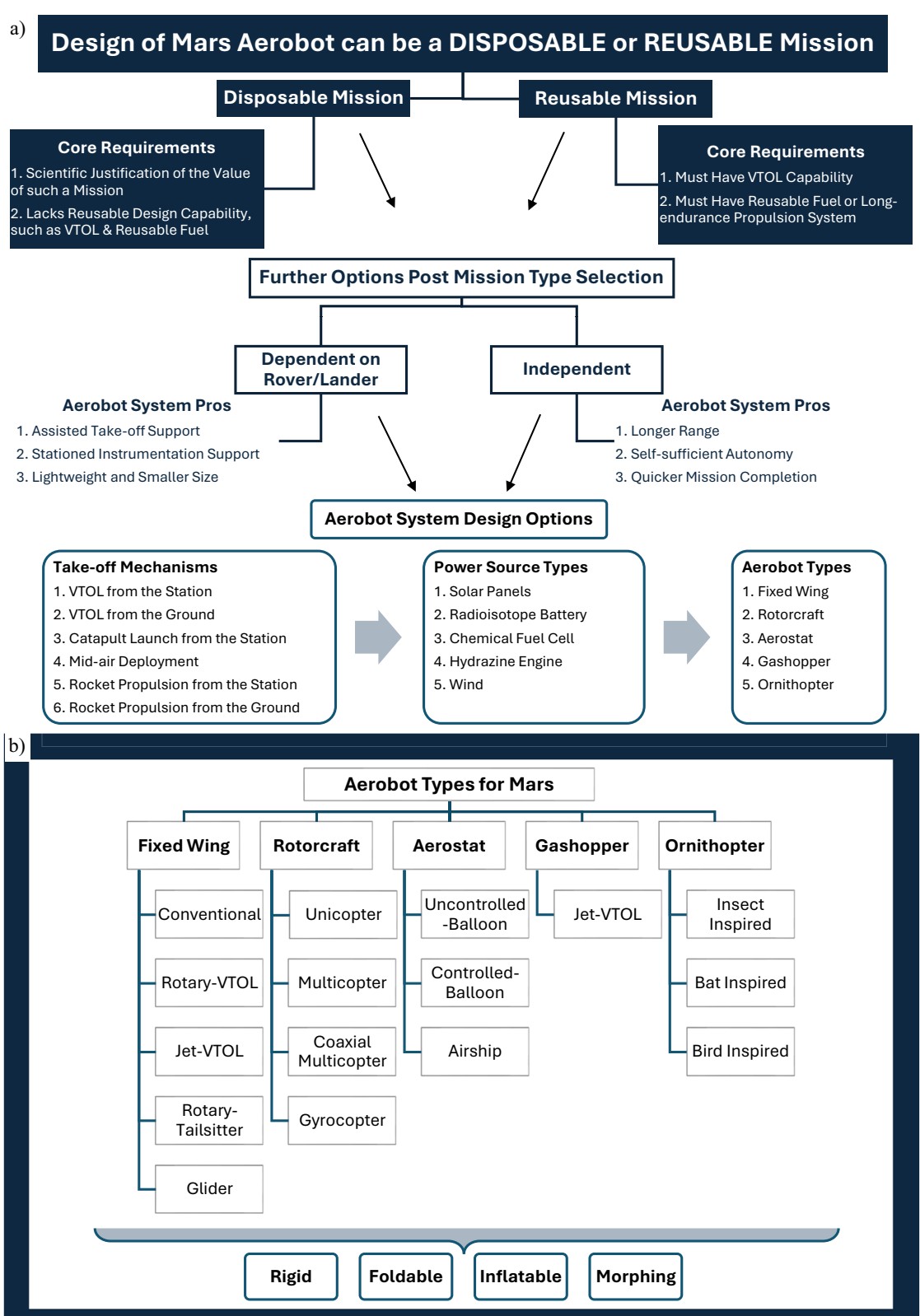

**Fig. 10 | Mars Aerobot Design Thinking Matrix linking mission context to system-level design choices. a** Mission-to-system decision flow linking mission type and environment to key subsystem choices. **b** Integrated classification of Mars aerobot types and structural configurations, synthesising concepts proposed in the literature.

that is not physically accessible by humans[122]. However, hydrazine's high toxicity requires careful handling, as demonstrated in one of the early concepts, the fixed-wing aircraft called Mini-Sniffer (Fig. 11b), studied by NASA's Dryden Research Center in 1977[120,122].

Alternatively, some design concepts, such as the Japanese Mars Airplane[123] and BIG BLUE inflatable-wing aircraft[124], proposed to utilise pre-charged electric batteries to power aerobots for a limited duration suitable for disposable missions.

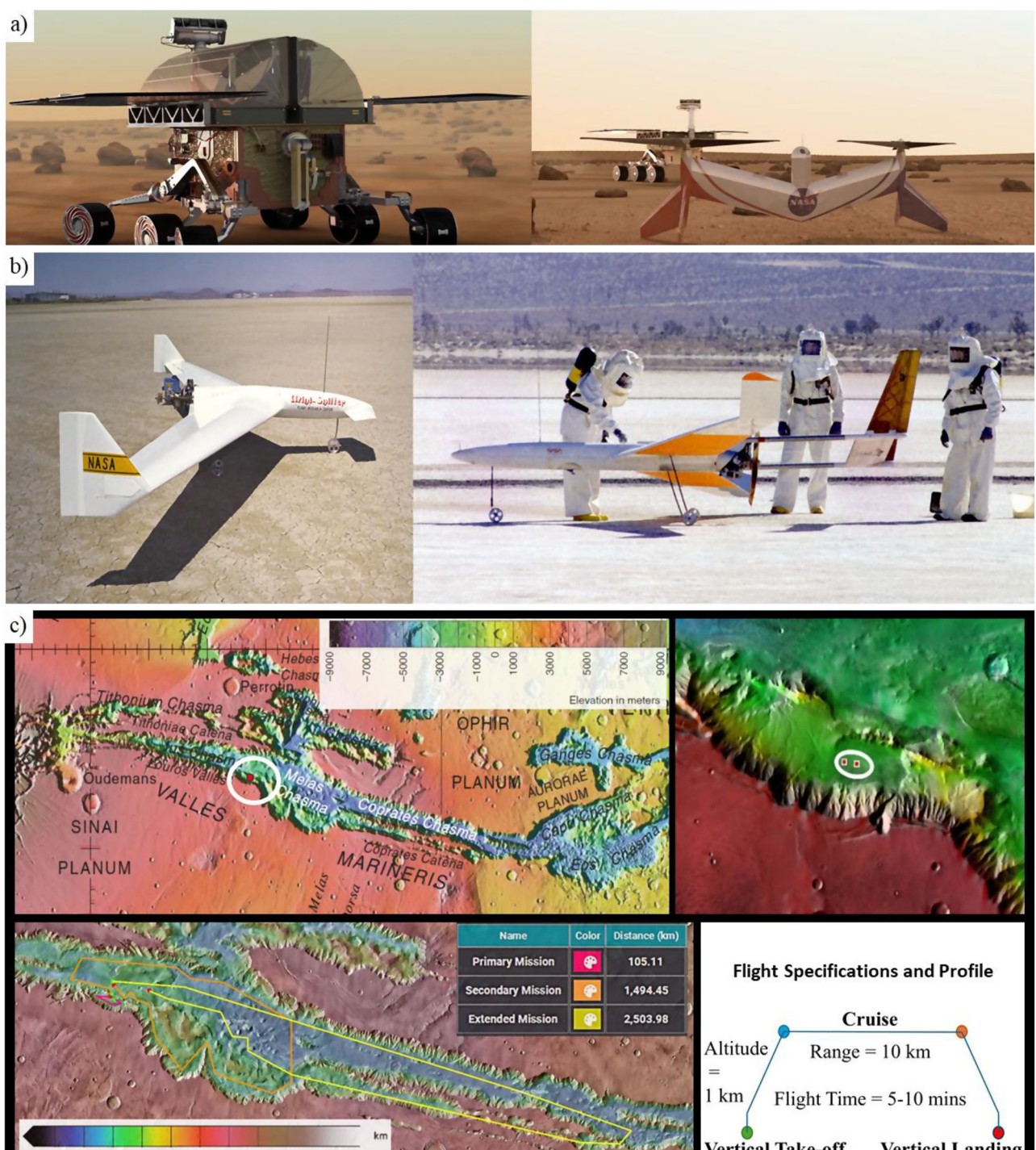

**Fig. 11 | Example Mars aerial concepts and an illustrative mission-level application of the framework. a** NASA's Mars Electric Reusable Flyer concept: a rover-dependent tail-sitter rotorcraft deployed by a rover-mounted robotic arm to explore caves and canyons, returning to the rover for recharging. Still images from a NASA Langley Research Center YouTube video[118] (public domain). **b** Mini-Sniffer III. The image on the right shows the crew wearing self-contained suits and oxygen tanks because the engine was fuelled with toxic hydrazine fuel. Image Credit: NASA

(public domain). **c** Panels adapted from the companion case study[36] (CC BY 4.0) showing a simulated mission in southwest Melas Basin: regional context within Valles Marineris, a close-up of the basin with two regions of interest and a representative landing ellipse[37], a mission-profile map with staged sorties, and an example flight specification/profile used to define sortie requirements. Base topography (public domain): USGS MOLA; GIS rendering: JMARS 5 (credit: NASA/JPL-Caltech/UArizona)[48].

In the recent literature, solar panels in combination with rechargeable batteries have been a popular choice of Martian aerobot researchers, with pioneering concepts such as solar-powered long-endurance Mars aircraft[125] and Martian Airborne Exploration Vehicle (MAEV)[126], because of the abundant renewable solar energy on the planet that makes the drone self-sustaining for a reusable mission type. Solar panels are lighter in weight when compared to the other power system options, and do not essentially require dedicated storage space in the fuselage, as they can embed in the top surfaces of the plane, particularly in winged aircraft. Over time, the panels get covered with dust and need cleaning for optimal efficiency, but unlike

Earth, it does not rain on Mars and the winds are inefficient in wiping off the stubborn layer closest to the skin, which causes the land spacecraft to have a limited life expectancy. Whereas, planetary aircraft would have a comparatively longer life because their high-speed flight works like a self-cleaning mechanism for the panels; however, global dust storms can be a terminator of spacecraft by blocking the sunlight for months from approaching the solar panels, as such in the case of Opportunity Rover's demise[28]. Moreover, solar energy is not fully accessible in the polar regions of Mars or areas that are hidden from sunlight (such as lava tubes), and the solar flux has both diurnal and seasonal variations.

For greater reliability, a radioisotope battery is a considerable but highly expensive option as a power source. Plutonium (Pu) 238 fuel-based radioisotope thermoelectric generator (RTG) has been used in several deep space missions, including the planetary Perseverance rover which carries a 4.8 kg Pu in a 45 kg multi-mission-RTG system with an ability to produce 110 Watt of electricity at a life expectancy of 14 years[104]. There are some preliminary Martian aerobot concepts in the literature that have opted for a radioisotope battery, such as Ballistic Mars Hopper[127], radioisotope-powered long-endurance Mars aircraft[125] and ARMaDA (Advanced Reconnaissance Martian Deployable Aircraft)[128]. An advanced alternative to the RTG is the Advanced Stirling Radioisotope Generator (ASRG), which, though still in NASA's testing phase, is proposed to produce several times more power with the same amount of radioisotope fuel and is being considered for future deep space missions, including the Mars Geyser Hopper[129]. However, these systems have significant drawbacks, including the heavier power system, the limited availability of nuclear fuel, and stringent regulations surrounding nuclear energy. Incorporating both a radioisotope battery and a solar power system on an independent, reusable aerobot could address certain mission requirements but at the cost of increased mass, size, propulsion power, and overall system complexity. This approach is more suitable for heavier aerobots, such as balloons and hoppers like the Mars Aerial Platform (MAP) balloon[130] and Mars Geyser Hopper[129]. Alternatively, an aerobot accompanied by a lander or rover equipped with a radioisotope battery can be used as an optional recharging station.

Another power source introduced in the Martian aerobot literature is fuel cells, an electrochemical device that converts the chemical energy of a reaction directly into electrical energy. Particularly, proton exchange membrane (PEM) fuel cells with oxygen and hydrogen gas as a fuel stored at cryogenic temperature and supercritical pressure, are chosen for use in the cold Martian environment because of their low-temperature operation, faster start-ups, swift response to changes in power demand, and high-power density with a lower weight and volume than other fuel cell types. An aerobot equipped with a fuel cell system would be transported to Mars with a limited fuel supply, making the mission either a disposable or short-term reusable type, as demonstrated in the MARV (Martian Autonomous Rotary-Wing Vehicle) concept[131] and MIRAGE (Mars Intelligent Reconnaissance Aerial and Ground Explorer)[132]. Continuous operation could only be achieved if a regenerative fuel cell (RFC) is used in combination with a solar power system, as considered in concepts like GTMARS (Georgia Tech Martian Autonomous Rotorcraft System)[133] and Mars Airship-Rover System[134]. The RFC system utilises electricity that is generated and stored diurnally via solar energy to dissociate liquid water, gathered as a by-product during the electricity generation process, back into gaseous hydrogen and oxygen through water electrolysis, thus creating a closed-loop system. RFC can be a superior electricity storage unit alternative to batteries for solar electric power on Mars due to its high specific energy, storage capacity, and power delivery, as it is deemed that longer discharge periods (such as nocturnal) along with higher energy storage requirements add a huge penalty to the battery mass and complexity[135]. Alternatively, a solid oxide fuel cell (SOFC) can generate power from carbon monoxide and oxygen, using Mars' $CO_2$, making it ideal for long-term missions that minimize reliance on Earth-supplied resources. However, SOFCs operate at over 800 °C, requiring complex thermal management and added system mass, which likely confines them to lander-based applications[88,135].

Power generation using wind turbines has not been widely considered for Martian aerobots, though studies are exploring wind turbines to complement solar power for human settlements, especially during global dust storms when winds are high but sunlight is minimal. Wind turbines alone cannot generate sufficient energy in Mars' thin atmosphere, and ground-based systems would require massive structures, such as a 50-m hub with a rotor diameter of over 80 m[136]. Modern airborne wind energy systems are proposed as a better alternative[136]; these systems involve tethered airborne devices, such as kites or drones, that capture wind energy at higher altitudes where winds are stronger and more consistent. Aerobots could either serve as part of these systems or assist in their deployment and maintenance. Alternatively, NASA is reviewing a smaller, foldable turbine with 2-m diameter blades, called the Planet Turbine Anemometer (PTA)[137]. Although wind energy harvesting on Mars is still an underdeveloped field, it has the potential to recharge Martian aerobots, particularly in polar regions and during global dust storms.

**Aerobot types for Mars**. The final step in the Mars Aerobot Design Thinking Matrix involves deciding on the type of aerobot most suitable for the specific Mars mission. The Fig. 10b illustrates a comprehensive categorisation of aerobot types, showcasing the various aerial vehicle designs proposed in the literature to fly on Mars. The classification is divided into five main categories: Fixed Wing, Rotorcraft, Aerostat, Gashopper, and Ornithopter, each representing a distinct approach to overcoming the unique challenges posed by Mars' thin atmosphere and harsh environmental conditions.

Fixed-wing is the oldest and most commonly proposed design, with many variants in the list of conceptual aerobots for Mars. Fixed-wing aerobots rely on traditional winged structures to generate lift. This category includes subtypes such as Conventional, Rotary-VTOL, Jet-VTOL, Rotary-Tailsitter, and Glider, highlighting the versatility within fixed-wing configurations. Conventional designs might be suited for long, sustained flights, while VTOL (Vertical Take-Off and Landing) variants combine the benefits of fixed wings with the ability to take off and land vertically, crucial in Mars' varied terrain. Gliders, which do not rely on continuous propulsion, could exploit Mars' atmospheric conditions for energy-efficient flight over extended distances.

Rotorcraft aerobots, including Unicopter, Multicopter, Coaxial Multicopter, and Gyrocopter, are characterised by their rotating blades, which provide lift and thrust. These designs offer superior manoeuvrability and the ability to hover, making them ideal for detailed surface exploration and navigation through complex landscapes. Multicopter designs, with their multiple rotors, provide enhanced stability and control, crucial for precise operations in Mars' unpredictable wind patterns.

Aerostats are lighter-than-air vehicles that rely on buoyancy rather than aerodynamic lift. This category includes Uncontrolled Balloons, Controlled Balloons, and Airships. These vehicles are particularly suited for long-duration missions, where energy efficiency and the ability to remain aloft for extended periods are paramount. Controlled balloons and airships can navigate more effectively, allowing for targeted exploration of specific areas on Mars, such as canyons or craters.

The Gashopper category is represented by Jet-VTOL or ballistic hopping designs, which are capable of vertical take-off and landing using jet propulsion. These aerobots could potentially use Martian resources, such as carbon dioxide from the atmosphere, to generate thrust. This type of vehicle could enable quick hops from one location to another, facilitating rapid exploration of Mars' diverse terrain.

Ornithopters are bio-inspired designs that mimic the flapping wings of birds, bats, or insects. Subtypes include Insect-Inspired, Bat-Inspired, and Bird-Inspired designs. These vehicles could take advantage of Mars' thin atmosphere by using flexible wings that adapt to changing conditions, offering a unique approach to flight that combines agility with energy efficiency. Ornithopters could be particularly useful for navigating through narrow spaces or for missions requiring a combination of hovering and gliding.

Beneath these categories, the chart (Fig. 10b) also highlights potential structural configurations of these aerobots: Rigid, Foldable, Inflatable, and Morphing. Rigid designs provide structural stability, Foldable designs offer compact storage and deployment capabilities, Inflatable designs can expand upon reaching Mars, reducing the space required during transport, and Morphing structures can dynamically change shape to adapt to varying flight conditions or mission requirements. This flexibility in design allows for a tailored approach to addressing the specific challenges of Mars exploration, enabling a wide range of mission types and operational strategies.

Overall, the diversity of potential aerobot designs underscores the importance of selecting the right configuration to meet the specific demands of Mars' environment, whether for detailed surface mapping, atmospheric sampling, or long-distance exploration. Each type offers unique advantages, and the choice of design will depend on the mission objectives, the specific environmental challenges, and the resources available for the mission.

*A comprehensive catalogue of historical, current, and proposed Martian aerobot concepts, including their mission objectives, technical specifications, and development status, is provided in the* Supplementary Data.

**Illustrative case study application of the design framework**. While this Perspective addresses the overarching planetary context and systems-engineering framework, a companion quantitative case study by the authors[36] illustrates its use by applying the underlying design logic to a rotary VTOL aerobot concept for a simulated Martian sample-return support mission, covering site selection, mission profile, flight operations, configuration trade-offs and preliminary sizing. The study grounds these steps in the SW Melas Basin within Valles Marineris, where regions of interest and a representative landing ellipse translate local terrain into a staged mission structure and flight requirements (Fig. 11c). Using the Mars Aerobot Design Thinking Matrix, the mission is framed as a reusable, rover-independent solar–battery VTOL system, narrowed to rotorcraft candidates screened via a momentum-theory trade study that shortlists a multicopter for preliminary sizing; the final configuration is then shaped by aeroshell packaging and deployment constraints, with a rigid architecture preferred to limit mechanical complexity and preserve power efficiency[36,138].

## Conclusion

Mars exploration is a critical scientific frontier, and aerobots are positioned as enabling assets for expanding in situ investigation under the planet's thin atmosphere, large thermal excursions, variable winds, and pervasive dust. The manuscript brings together planetary context, areography, and climatology with engineering considerations to outline a coherent pathway from mission intent to feasible aerial systems. It emphasises the need to align lift generation, energy management, thermal control, structural robustness, and autonomy with site selection, seasonal windows, and operational constraints. The integrated framework, captured in the Mars Aerobot Design Thinking Matrix, highlights how orders of relief, atmospheric structure (including boundary-layer behaviour), and dust activity shape vehicle configuration, deployment, and flight envelopes. It underscores the value of design sequencing such as defining mission type, take-off mechanism, power architecture, and aerobot class before committing to detailed trades; so that packaging, mass, and environmental margins remain consistent throughout development. Ingenuity's demonstrations motivate disciplined advances in rotor aerodynamics, structural stiffness, sensing and navigation, and ground-test methods that reflect low-density, low-damping regimes.

The paper identifies priorities for future work: improved aerodynamic performance at low Reynolds number; propulsion and energy strategies resilient to dust and irradiance variability; materials and coatings tolerant of thermal cycling and abrasion; autonomy capable of reliable navigation without GNSS; and delivery approaches that reduce dependence on flat, hazard-free terrain. Collectively, these developments support safe, repeatable flights, higher-fidelity atmospheric measurements within the planetary boundary layer, and targeted access to scientifically compelling terrains. The framework is intended to help researchers and engineers maintain traceability from environmental drivers to testable requirements, thereby improving mission reliability and scientific return while supporting broader programmatic goals for Mars exploration.

## Data availability

Data sharing not applicable to this article as no new datasets were generated or analysed. This Perspective draws on publicly available mission data and published literature, which are cited in the manuscript and Supplementary Information. The aerobot catalogue compiled from published sources is provided as Supplementary Data.

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

## Acknowledgements

The authors thank NASA and the wider Mars exploration and planetary robotics community for making mission data, imagery, and technical resources openly available, and for the broader culture of knowledge sharing that informed this Perspective and framework.

## Author contributions

Vishal Youhanna conceived the study, developed the framework and design matrix, conducted the literature synthesis and analysis, and wrote the manuscript. Leonard Felicetti and Dmitry Ignatyev provided academic supervision and reviewed the manuscript.

## Competing interests

The authors declare no competing interests.
