## [Transparent Peer Review file · Communications Engineering]

Mars Planetary Insights and Design Framework for Future In-Situ Aerial Robotic Missions

Corresponding Author: Dr Vishal Youhanna

Version 1:

Reviewer comments:

Reviewer #1

(Remarks to the Author)

The paper is very interesting giving a general view of the process in selecting an aerial system for Mars exploration giving good insights in Mars' environment and trying to highlight the challenges that a similar mission must face.

The introduction is very well developed and sufficient details are given to the reader.

Very complete background and framework for the analysis is presented and all necessary justifications are shown.

I think it was not in the scope of the paper but a possible comparison through a simulated mission could increase the details necessary for a final decision on the specific mission.

English is good.

Reviewer #2

(Remarks to the Author)

Review of Youhanna, V, Ignatyev, D and Felicetti, L (2025) "Mars Planetary Insights and Design Framework for Future In-Situ Aerial Robotic Missions"

The paper presents a framework for designing and deploying aerobots on Mars considering the harsh Martian environment and presents a rationale for doing so in the context of the desired mission goals, given the science questions most suitably addressed by such vehicles.

After the Introduction, the paper begins by setting background context with a very good summary of the history and findings of previous Mars missions.

I have a few minor points for section 2:

Section 2

Minor: Page 3, para 2, line 1: "two-fifth" is usually expressed as "two-fifths" in English – but this may be American usage where plurals are often dropped (e.g. math/math – mathematics is a set of subjects not a single one!).

Page 3, para. 3, line 7: "...Inhabiting Mars life..." – should be "...inhabited by Mars life...".

Page 4 – I don't see a reference to Figure 2.3 in the text. This should perhaps come after "Mars is renowned for dust storms" (as shown in Fig. 2.3).

Suggestion: Page 4 – Discussing the Soviet Mars 3 mission, it may be noted that this mission may have transmitted the first image of the surface of Mars taken from the surface – sadly the image is not clear enough to definitively say it was the horizon that was imaged. See:

<https://www.planetary.org/articles/0412-how-we-searched-for-mars-3>

Suggestion: Page 5, line 3 – perhaps "Mars' surface from the top of its atmosphere" is better than "Mars ground from its top atmosphere".

Page 5, line 7 – please spell out "CNSA" on first usage – China National Space Administration (CNSA).

Page 5, para 2, line 2 – "...which have been manageable...the most." "...which have been able to be managed practically, thus enabling its surface to be explored more widely and with more success."

Same line – "Viking 1, the first successful Mars lander..." "Viking 1, the first truly successful Mars lander..." - the Soviets might argue that Mars 3 was a successful lander albeit for a short time.

Page 5, para 2, line 7 – "...such as the rover.... and the helicopter...." [definite articles]

Page 5, para 2, line 12 – "...Curiosity rover, comparatively, has covered...."

Page 5, para 2, line 13 – “Relatively, Martian spacecraft are slow...” [“spacecraft” is self-plural like “sheep”]
“...but well achievers for planetary exploration.” “... but have been astonishingly successful as planetary explorers.”
Page 5, para 3, line 4 – “...before getting damaged...” “... before being damaged...”
Page 5, para 3, line 5 – “...next generation aerobot...” “... next generation of aerobot...”

Section 3

Section 3 covers the impact of the terrain, atmosphere and radiation environments of the aircraft and its systems. Again, the paper summarises the issues well, although would benefit from some extra detail in places. In particular, some assertions need quantitative analyses (or references to such) to support them.

General comment:

A lot of numerical information is given within the text – perhaps some of this would be more clearly presented in a table. For example, there is a good comparative summary of Mars vs. Earth’s atmosphere and its characteristics vis-a-vis aeronautics in Table 1 of:

IAC-17-A3.3A.10 Design and Control of a Y-4 Tilt-Rotor VTOL Aerobot for Flight on Mars

<https://openresearch.surrey.ac.uk/esploro/outputs/conferencePresentation/Design-and-Control-of-a-Y-4/99511131102346>

Section 3.1, para. 1, line 1 – “...aircrafts...” should be “aircraft” – like spacecraft – all self-plural.

Section 3.1, para. 1, line 2 – “...recharging their power source...”

Section 3.1.1, line 2 – “with a ratio of 1 to 2” – what has a ratio 1 to 2 – the surface area? [I think this is what you mean, but be explicit]

Section 3.1.1, line 5 – You might mention that the zero-elevation datum level in the map is taken as where the surface pressure is 6.2 mbar (the atmospheric pressure triple point of water on Mars). This gives the context for aircraft design as for Earth the equivalent is 1013 mbar.

Section 3.1.1, line 9 – there is an empty pair of brackets () – I think perhaps you meant to refer to Fig. 3.2 here – otherwise Fig. 3.2 is not referenced.

Page 8, line 1 – The United States Geological Survey (USGS)...

Section 3.1.1, page 8, last para. – indeed, the elevation etc. will affect the aerodynamic requirements of the aerobot – perhaps it would be useful to enumerate this more quantitatively by means of a graph of pressure, density and temperature vs altitude (above datum).

Such a graph can be found here:

https://www.researchgate.net/figure/Mars-atmospheric-pressure-temperature-and-density-as-functions-of-the-altitude-relative_fig8_330210450

It summarises neatly all the key information an aerobot designer needs with respect to atmospheric conditions vs. altitude. Note the temperature inversion which has implications for the speed of sound vs. altitude.

Section 3.1.2 line 4 – “aspects of this hierarchical classification are shown in Fig. 3.4”

Section 3.1.2 line 5 – “For aerobot design, the fourth-order...”

You state that Fifth-order regional features pose less of a challenge.

I don’t quite agree with that statement – they pose different challenges – especially to the undercarriage design.

This has to be thought through very carefully to design a system that can robustly sit on the surface correctly, and not be affected by rocks, boulders etc. It also requires the aerobot to have good hazard detection systems such as all-round machine vision, so that it can autonomously land safely (Earth flight control not being possible due to the speed of light delay).

This is not a trivial problem for any large aerobot. NASA’s Ingenuity – being very small – was not so badly affected, as it could more easily find the required space between rocks.

Assuming VTOL. Should the undercarriage be stalk-like? 3 or 4 stalks? (3 will always sit on a plane – 4 won’t necessarily).

Should pads be used to stop the aerobot sinking into soft terrain – if so, how big?

Should skis be used instead, as for many terrestrial helicopters?

How high should the undercarriage be to avoid rocks under the body?

What are the implications for vehicle stability given the Martian winds?

Should the undercarriage be retractable or not?

What is the effect of dust on such mechanisms?

Should the aerobot use a rocker-arm mobility system similar to that of the rovers? Does it need land mobility? – these are all design choices for the aerobot designer.

Page 9, line 3 – Other distinctive features, the Medusae...

[Latin feature names are usually italicised – Medusae Fossae, etc. but check Nature’s in-house style.]

Section 3.1.3, line 2 – “...free-air gravity map (120 km resolution (Fig. 3.6) charts these...”

You state that variations in gravity are significant enough to impact the design.

Comparing the force variation to other force variability, e.g. weather, wind loading, flight altitude variation, etc. – just how big a factor is gravity variability – can you please quantify this?

In my view a 1% variation in weight is well within the capabilities of any aircraft control system – indeed the weight may not be known to this precision, given dust loading etc.

A variability of 0.03 N/kg doesn’t seem to be much compared, for example, to the aerodynamic forces (drag, lift) and their variability. Consider the weight and drag increase due to dust etc.

The GMM-3 map only has a resolution of 120 km – this seems very large for precise path planning, hazard avoidance, subsurface feature detection etc. you describe.

I think this section needs a more quantitative analysis to support your contention that gravity anomalies are significant. I believe these effects would be “in the noise” given the many other effects of the Martian environment on the aerodynamic efficiency and control requirements of a specific vehicle and a specific flight path.

Please quantitatively justify these assertions or moderate this section to indicate that the gravity field may be a factor that needs consideration if the aircraft is acting at its performance limits or has a particularly sensitive control system. Or simply remove it.

As you state at the end of Section 3.1.2 – the atmosphere (and its physical properties, weather, winds and dust storms, etc.) is a key driver for the aerobot design, and in my view, is the dominant one.

Page 12 – surely the weather phenomena associated with the seasonal mass exchanges are more critical than minor temporal gravity variations as far as vehicle aerodynamics are concerned? Winds and seasonal dust storms have the potential for a major impact on the safety and viability of aerobots over the longer term.

Section 3.2.2., para. 1 would be a good place to insert the atmospheric profile graphs mentioned above.

Section 3.2.2.2., line 5 – ...it mobilises...

Section 3.2.2.3, para 2. – here you mention Reynold's Number which is indeed a key parameter. You might mention that the Reynold's number for typical Mars aircraft is small compared to the equivalent on Earth – for that reason the flight regime is more reminiscent of insect flight than of traditional terrestrial aircraft design. This requires some re-thinking of design approaches.

Section 3.2.3, line 1 – "...on a daily frequency"

Page 15, line 7 – the word "spacecraft" should be "aircraft" or "Mars aircraft".

[Yes indeed, as with any spacecraft – charging and ESD mitigation and ionising radiation effects mitigation will be necessary on Mars aerobots.]

Section 3.2.4, line 1. – Mars is subject to significant ionising particle..."

Section 3.2.4 – the supplementary notes S1 add useful detail

Section 3.2.5 – the supplementary notes S2 add useful detail.

Section 4

Section 4 is an excellent and well written summary of the design and testing of the Ingenuity aerobot and the implications for future aerobot design.

Section 4, line 5 – define GPS. (without navigational information provided by systems such as the Global Positioning System (GPS) orbiting Earth.

Section 4, line 8 – "The Ingenuity ..."

This section also refers to gravity anomalies as being an issue – please see previous comments. I think local weather may be a bigger problem.

Section 4.1.1.1. line 1 "...was conducted in the NASA Jet Propulsion Laboratory (NASA/JPL)..."

Section 5

Section 5 introduces the main contribution of the paper – a methodology for thinking about Mars aerobot design. It is an exemplar of the systems design principles which are (or should be) applied to all space missions. This is an excellent section.

Page 23, para 3, line 6 "2000 RPMs" is a very specific figure and is highly dependent upon the nature of the aerobot. An aerostat, for example, has no need to generate lift this way, of a hybrid aerobot (see the IAC paper referred to above), may have very different requirements.

This should be deleted and replaced by

"...due to the requirement of generating sufficient thrust – and is especially an issue for a helicopter style of aerobot, such as Ingenuity, which has to generate lift aerodynamically via rotor blades which in turn requires rotor tip speeds close to the speed of sound on Mars; therefore, Ingenuity has a flying endurance of only..."

Page 27, para 1, line 7 refers to "aerial spacecrafts" – these are aircraft! A spacecraft that operates in a planetary atmosphere is still an aircraft.

Please note throughout the paper "spacecraft" and "aircraft" should be used for the plural.

Page 27, para 1, line 3 Perseverance is not a deep space spacecraft – its host carrier was but this is a planetary rover – as such it is not operating in space (by definition a vacuum).

RTGs have been used on deep space missions such as Voyager, etc., where the use of solar power is impractical. It has also been used on lunar and planetary surfaces, such as in the Perseverance rover, where a long-term consistent power source, independent of sunlight, has been required.

Section 6

Fine.

Overall, this is a very good paper which introduces the reader to many of the key issues involved in Mars aerobot design – no easy task. The paper provides a good background and a clear exposition of the issues. The use of real and proposed missions from literature illustrates the work very well.

The paper is novel in that it brings all the issues together in one place and offers a structured "systems design" approach to thinking about the design problem.

The paper will be of value to any researcher embarking on Mars aerobot design.

I recommend publication, subject to the corrections outlined in my review.

My comments and mainly regarding issues of English and clarity.

My only significant technical issue comes with the section of the small (1%) variability in gravity measured from orbit over large surface areas – I do not believe this is a significant issue for aerobot design given that there are many other sources of force variability acting on the aircraft.

I remain to be convinced – but that requires a proper comparison of the variability of all the forces and their variability acting on the aerobot.

In the absence of such a quantitative analysis, I would be happy if the section was moderated as I have suggested to say that gravitation variability may need to be taken into account for aircraft acting close to their performance or control limits.

Prof Craig Underwood

Emeritus Professor of Spacecraft Engineering,
Surrey Space Centre,
University of Surrey,
Surrey, GU2 7XH
United Kingdom

Version 2:

Reviewer comments:

Reviewer #1

(Remarks to the Author)

I would like to thank the authors for revising their article, taking into account the reviewers' requests. The article is now more comprehensible and coherent. I believe it can be published with the changes made.

Reviewer #2

(Remarks to the Author)

Thank you for your responses to my comments on the original submission.

I am satisfied with the responses and I am now happy for the paper to go ahead to publication.

Congratulations on producing a good and useful paper for future Mars' exploration.

Response to Reviewers

Nature Communications Engineering

Subject: Resubmission of Revised (R1) Manuscript COMMS-25-0310A: "*Mars Planetary Insights and Design Framework for Future In-Situ Aerial Robotic Missions*"

We thank the reviewers for their detailed and constructive feedback, which has helped us clarify the scope and strengthen the contribution of this Perspective. We have addressed all substantive points raised and made targeted revisions throughout the manuscript.

All changes are highlighted in the revised manuscript, including a small number of minor edits made for clarity and consistency beyond the specific reviewer comments. In line with the editor's guidance, we also streamlined the manuscript and consolidated figures into multi-panel layouts, reducing the total to 11 figures while preserving legibility.

Below we provide a point-by-point response to the reviewers' comments and indicate the corresponding changes made in the manuscript.

Reviewer #1

Comment: "I think it was not in the scope of the paper but a possible comparison through a simulated mission could increase the details necessary for a final decision on the specific mission."

Response: We thank the reviewer for this valuable suggestion. We agree that a simulated mission comparison is an important step toward supporting a final decision on a specific mission concept. A detailed mission-level application of the design logic has already been carried out in our previous work (Reference [36]), where we apply the methodology to a representative Martian rotorcraft mission, including mission profile definition, configuration trade-offs, and power/mass estimation. To avoid duplicating this material and in view of the journal's word and figure limits, we have not reproduced the full case study here. Instead, in the revised manuscript we explicitly highlight [36] as the quantitative worked example that demonstrates how the framework can be applied in practice (see Section 5.2.4 Illustrative Case Study Application of the Design Framework). All changes are highlighted in the revised manuscript.

Reviewer #2

We thank the reviewer for their careful reading of the manuscript and for the constructive comments, which have helped improve clarity and precision. All changes are highlighted in the revised manuscript.

Please note that page and line numbers referenced by the reviewer refer to the original submission; they may have shifted in the revised manuscript due to text edits and figure consolidation.

Section 2

Grammatical phrasing and terminology comments:

Minor: Page 3, para 2, line 1: “two-fifth” is usually expressed as “two-fifths” in English – but this may be American usage where plurals are often dropped (e.g. math/maths – mathematics is a set of subjects not a single one!).

Page 3, para. 3, line 7: “...Inhabiting Mars life...” – should be “...inhabited by Mars life...”.

Page 4 – I don’t see a reference to Figure 2.3 in the text. This should perhaps come after “Mars is renowned for dust storms” (as shown in Fig. 2.3).

Suggestion: Page 5, line 3 – perhaps “Mars’ surface from the top of its atmosphere” is better than “Mars ground from its top atmosphere”.

Page 5, line 7 – please spell out “CNSA” on first usage – China National Space Administration (CNSA).

Page 5, para 2, line 2 – “...which have been manageable...the most.” “...which have been able to be managed practically, thus enabling its surface to be explored more widely and with more success.”

Same line – “Viking 1, the first successful Mars lander...” “Viking 1, the first truly successful Mars lander...” - the soviets might argue that Mars 3 was a successful lander albeit for a short time.

Page 5, para 2, line 7 – “...such as the rover... and the helicopter...” [definite articles]

Page 5, para 2, line 12 – “...Curiosity rover, comparatively, has covered...”

Page 5, para 2, line 13 – “Relatively, Martian spacecraft are slow...” [“spacecraft” is self-plural like “sheep”]

“...but well achievers for planetary exploration.” “... but have been astonishingly successful as planetary explorers.”

Page 5, para 3, line 4 – “...before getting damaged...” “... before being damaged...”

Page 5, para 3, line 5 – “...next generation aerobot...” “... next generation of aerobot...”

Response: All of the suggested grammatical, wording and terminology corrections in Section 2 have been implemented in the revised manuscript. Page and line numbers may have shifted slightly due to figure bundling, but the underlying sentences have been updated as indicated.

Other Comments:

Suggestion: Page 4 – Discussing the Soviet Mars 3 mission, it may be noted that this mission may have transmitted the first image of the surface of Mars taken from the surface – sadly the image is not clear enough to definitively say it was the horizon that was imaged. See:

<https://www.planetary.org/articles/0412-how-we-searched-for-mars-3>

Response: We thank the reviewer for this historical clarification. The sentence has been revised to note the brief image transmission from Mars 3.

Section 3

Grammatical phrasing and terminology comments:

Section 3.1, para. 1, line 1 – “...aircrafts...” should be “aircraft” – like spacecraft – all self-plural.

Section 3.1, para. 1, line 2 – “...recharging their power source...”

Section 3.1.1, line 2 – “with a ratio of 1 to 2” – what has a ratio 1 to 2 – the surface area? [I think this is what you mean, but be explicit]

Section 3.1.1, line 9 – there is an empty pair of brackets () – I think perhaps you meant to refer to Fig. 3.2 here – otherwise Fig. 3.2 is not referenced.

Page 8, line 1 – The United States Geological Survey (USGS)...

Page 9, line 3 – Other distinctive features, the Medusae...

[Latin feature names are usually italicised – *Medusae Fossae*, etc. but check Nature’s in-house style.]

Section 3.2.2.2., line 5 – ...it mobilises...

Section 3.2.3, line 1 – “...on a daily frequency”

Page 15, line 7 – the word “spacecraft” should be “aircraft” or “Mars aircraft”.

[Yes indeed, as with any spacecraft – charging and ESD mitigation and ionising radiation effects mitigation will be necessary on Mars aerobots.]

Response: We thank the reviewer for these careful suggestions. All of the suggested grammatical, wording and terminology corrections in Section 3 have been implemented in the revised manuscript. Page and line numbers may have shifted slightly due to figure bundling, but the underlying sentences have been updated as indicated.

General comment:

A lot of numerical information is given within the text – perhaps some of this would be more clearly presented in a table. For example, there is a good comparative summary of Mars vs. Earth’s atmosphere and its characteristics vis-a-vis aeronautics in Table 1 of:

IAC-17-A3.3A.10: Design and Control of a Y-4 Tilt-Rotor VTOL Aerobot for Flight on Mars

Response: We appreciate the suggestion to summarise the numerical comparisons in tabular form and agree that Table 1 in the cited work provides a clear illustration of the key contrasts between Earth and Mars. In this Perspective, however, we have chosen to keep these values in the narrative rather than introduce an additional table, in order to stay within the journal’s length and figure constraints and to maintain the flow of the planetary-to-design discussion. We therefore have not added a new table or reproduced the suggested one, but we believe the main quantitative contrasts are already conveyed by the existing text.

Other comments:

Section 3.1.1, line 5 – “You might mention that the zero-elevation datum level in the map is taken as where the surface pressure is 6.2 mbar (the atmospheric pressure triple point of water on Mars). This gives the context for aircraft design as for Earth the equivalent is 1013 mbar.”

Response: We appreciate the motivation of this suggestion; however, because the current MOLA areoid is formally defined as a gravitational–rotational equipotential rather than by a fixed pressure surface, we have chosen to retain a more general description that emphasises its topographic and paleo-oceanic context without specifying a particular pressure value.

Section 3.1.1, page 8, last para – indeed, the elevation etc. will affect the aerodynamic requirements of the aerobot – perhaps it would be useful to enumerate this more quantitatively by means of a graph of pressure, density and temperature vs altitude (above datum).

Section 3.2.2., para. 1 – would be a good place to insert the atmospheric profile graphs mentioned above.

Response: We appreciate the reviewer for this helpful suggestion and agree that the variation of pressure, density and temperature with altitude is important for aerobot design. In the present Perspective, these quantitative trends are already treated in the Mars climatology discussion, with an emphasis on the lower-atmosphere regime relevant to aerobot operations, rather than the full entry-to-space column. In order to remain within the journal's figure limits and avoid duplicating material, we have not added an additional altitude-profile figure at this stage, but we have ensured that the text in Section 3.1.1 (last paragraph) clearly states that elevation affects pressure, density and temperature and thereby the aerodynamic and power requirements of the aerobot.

Section 3.1.2, line 5 –You state that Fifth-order regional features pose less of a challenge. I don't quite agree with that statement – they pose different challenges – especially to the undercarriage design....

Response: We thank the reviewer for this thoughtful comment. To avoid any suggestion that fifth-order roughness poses a lesser challenge, we have revised the text in Section 3.1.2 to remove that wording and to state explicitly that small-scale slopes, rocks and soft deposits introduce critical landing and ground-interaction challenges for VTOL aerobots.

Our aim was to emphasise that fifth-order roughness influences design at a different level of decision-making within the hierarchy, rather than being less important. In our use of the orders-of-relief framework, fourth-order regional features define the terrain provinces that primarily drive vehicle architecture and the baseline landing concept, while fifth-order features represent the local expression of roughness within those provinces and refine site-specific undercarriage and hazard-detection requirements.

Section 3.1.3 (gravity variability and GMM-3): The reviewer questions whether ~1% variability in gravity is significant for aerobot design and notes that other force variations (atmosphere, winds, etc.) are likely dominant. In the absence of a full quantitative comparison, they suggest moderating the section to state that gravitational variability may only need to be considered when aircraft operate close to performance or control limits.

Response: We thank the reviewer for this careful assessment. Upon consideration, we agree with the reviewer that the ~1% spatial variability in Mars' gravitational acceleration is small compared with variations in atmospheric density, winds and dust loading, and that it is therefore a secondary consideration for most low-altitude aerobots. In Section 3.1.3 we have now explicitly stated this and revised the text to remove any implication that gravity anomalies are a primary design driver. We instead describe local gravity as mainly relevant in two contexts: (i) performance and control analysis for vehicles operating close to their limits, and (ii) high-fidelity trajectory and EDL modelling at mission level. In the latter case we now cite the NASA JPL-LLIS lesson [54], which documents how simplified gravity assumptions contributed to landing-condition errors on a past Mars mission. We have also removed earlier statements suggesting roles in hazard avoidance or detailed path planning, so that the discussion is consistent with the reviewer's recommendation.

Section 3.2.2.3, para 2. – here you mention Reynold's Number which is indeed a key parameter. You might mention that the Reynold's number for typical Mars aircraft is small compared to the equivalent on Earth – for that reason the flight regime is more reminiscent of insect flight than of traditional terrestrial aircraft design. This requires some re-thinking of design approaches.

Response: We appreciate this insightful comment. We have added a brief statement in Section 3.2.2.3 noting that Reynolds numbers on Mars are typically lower than terrestrial

equivalents, which can shift the aerodynamic regime toward terrestrial insect-flight conditions and motivate adapted design approaches.

Section 4

Grammatical phrasing and terminology comments:

Section 4, line 5 – "define GPS. (without navigational information provided by systems such as the Global Positioning System (GPS) orbiting Earth.

Section 4, line 8 – "The Ingenuity"

Section 4.1.1.1, line 1 "...was conducted in the NASA Jet Propulsion Laboratory (NASA/JPL)..."

Response: We thank the reviewer for these careful suggestions. All of the suggested grammatical, wording and terminology corrections in Section 4 have been implemented in the revised manuscript. Page and line numbers may have shifted slightly due to figure bundling, but the underlying sentences have been updated as indicated.

Other comments:

Section 4, line 8 – gravity anomalies – This section refers again to gravity anomalies as being an issue; I think local weather may be a bigger problem.

Response: We thank the reviewer for this clarification and agree that local atmospheric conditions are the dominant driver for aerobot operations. In Section 4 Introduction, we have therefore removed the reference to gravity anomalies.

Section 5

Page 23, para 3 – "2000 RPMs": The reviewer notes that a specific rotor speed is vehicle-dependent and not applicable to all aerobots, and suggests a more general description linked to helicopter-style vehicles such as Ingenuity.

Response: We thank the reviewer for this helpful suggestion and agree that quoting "2000 RPMs" is too specific. We have removed this value and now describe the constraint more generally, stating that helicopter-type aerobots such as Ingenuity must generate sufficient thrust aerodynamically with rotor blades operating at high tip speeds close to the local speed of sound, which in turn limits Ingenuity's flight endurance to only a few minutes (Section 5.1)

Page 27, para 1, line 7 refers to "aerial spacecrafts" – these are aircraft! A spacecraft that operates in a planetary atmosphere is still an aircraft.

Response: We thank the reviewer for this terminology clarification. While NASA mission documentation sometimes refers to Ingenuity within a broader mission "spacecraft" context (Ingenuity Press-kit) and 'aerial spacecraft' appears in some NASA mission-taxonomy usage (NASA NTRS - 20170009572), we agree that "planetary aircraft" is the clearest terminology for this manuscript and have revised the text accordingly.

Page 27, para 1, line 3 Perseverance is not a deep space spacecraft – its host carrier was but this is a planetary rover – as such it is not operating in space (by definition a vacuum).

Response: We thank the reviewer for this clarification. We have revised the text to refer to "deep-space missions" rather than "deep-space spacecraft" and to clearly identify

Perseverance as a planetary rover that uses an RTG to provide long-term, sunlight-independent power on the Martian surface.

We believe these revisions significantly enhance the manuscript and address the reviewers' concerns.

Thank you for your consideration, and we look forward to your feedback.

Review of Youhanna, V, Ignatyev, D and Felicetti, L (2025) “Mars Planetary Insights and Design Framework for Future In-Situ Aerial Robotic Missions”

The paper presents a framework for designing and deploying aerobots on Mars considering the harsh Martian environment and presents a rationale for doing so in the context of the desired mission goals, given the science questions most suitably addressed by such vehicles.

After the Introduction, the paper begins by setting background context with a very good summary of the history and findings of previous Mars missions.

I have a few minor points for section 2:

Section 2

Minor: Page 3, para 2, line 1: “two-fifth” is usually expressed as “two-fifths” in English – but this may be American usage where plurals are often dropped (e.g. math/maths – mathematics is a set of subjects not a single one!).

Page 3, para. 3, line 7: “...Inhabiting Mars life...” – should be “...inhabited by Mars life...”.

Page 4 – I don’t see a reference to Figure 2.3 in the text. This should perhaps come after “Mars is renowned for dust storms” (as shown in Fig. 2.3).

Suggestion: Page 4 – Discussing the Soviet Mars 3 mission, it may be noted that this mission may have transmitted the first image of the surface of Mars taken from the surface – sadly the image is not clear enough to definitively say it was the horizon that was imaged. See:

<https://www.planetary.org/articles/0412-how-we-searched-for-mars-3>

Suggestion: Page 5, line 3 – perhaps “Mars’ surface from the top of its atmosphere” is better than “Mars ground from its top atmosphere”.

Page 5, line 7 – please spell out “CNSA” on first usage – China National Space Administration (CNSA).

Page 5, para 2, line 2 – “...which have been manageable...the most.” “...which have been able to be managed practically, thus enabling its surface to be explored more widely and with more success.”

Same line – “Viking 1, the first successful Mars lander...” “Viking 1, the first truly successful Mars lander...” - the soviets might argue that Mars 3 was a successful lander albeit for a short time.

Page 5, para 2, line 7 – “...such as the rover.... and the helicopter...” [definite articles]

Page 5, para 2, line 12 – “...Curiosity rover, comparatively, has covered...”

Page 5, para 2, line 13 – “Relatively, Martian spacecraft are slow...” [“spacecraft” is self-plural like “sheep”]

“...but well achievers for planetary exploration.” “... but have been astonishingly successful as planetary explorers.”

Page 5, para 3, line 4 – “...before getting damaged...” “... before being damaged...”

Page 5, para 3, line 5 – “...next generation aerobot...” “... next generation of aerobot...”

Section 3

Section 3 covers the impact of the terrain, atmosphere and radiation environments of the aircraft and its systems. Again, the paper summarises the issues well, although would benefit from some extra detail in places. In particular, some assertions need quantitative analyses (or references to such) to support them.

General comment:

A lot of numerical information is given within the text – perhaps some of this would be more clearly presented in a table. For example, there is a good comparative summary of Mars vs. Earth’s atmosphere and its characteristics vis-a-vis aeronautics in Table 1 of:

IAC-17-A3.3A.10 **Design and Control of a Y-4 Tilt-Rotor VTOL Aerobot for Flight on Mars**

<https://openresearch.surrey.ac.uk/esploro/outputs/conferencePresentation/Design-and-Control-of-a-Y-4/99511131102346>

	Earth at Sea Level	Mars
Gravity (m/s^2)	9.81	3.71
Atmospheric Composition	N ₂ - 78.08%	CO ₂ - 95.32%
	O ₂ - 20.95%	N ₂ - 2.7%
	H ₂ O - 0.4%	Ar - 1.6%
	Ar - 0.93%	O ₂ - 0.13%
	CO ₂ - 0.036%	CO - 0.08%
Atmospheric Density (kg/m^3)	1.225	0.0138
Average Temperature (K)	288.15	210.15
Average Wind Speeds (m/s^2)	0-100 where 14-17m/s is difficult to walk	2-7 (summer), 5-10 (fall), 17-30 (dust storm)
Speed of Sound (m/s^2)	340.3	245
Dynamic Viscosity (Ns/m^2)	1.789×10^{-5}	1.2235×10^{-4}

Section 3.1, para. 1, line 1 – “...aircrafts...” should be “aircraft” – like spacecraft – all self-plural.

Section 3.1, para. 1, line 2 – “...recharging their power source...”

Section 3.1.1, line 2 – “with a ratio of 1 to 2” – what has a ratio 1 to 2 – the surface area? [I think this is what you mean, but be explicit]

Section 3.1.1, line 5 – You might mention that the zero-elevation datum level in the map is taken as where the surface pressure is **6.2 mbar** (the atmospheric pressure triple point of water on Mars). This gives the context for aircraft design as for Earth the equivalent is **1013 mbar**.

Section 3.1.1, line 9 – there is an empty pair of brackets () – I think perhaps you meant to refer to Fig. 3.2 here – otherwise Fig. 3.2 is not referenced.

Page 8, line 1 – The United States Geological Survey (USGS)...

Section 3.1.1, page 8, last para. – indeed, the elevation etc. will affect the aerodynamic requirements of the aerobot – perhaps it would be useful to enumerate this more quantitatively by means of a graph of pressure, density and temperature vs altitude (above datum).

Such a graph can be found here:

https://www.researchgate.net/figure/Mars-atmospheric-pressure-temperature-and-density-as-functions-of-the-altitude-relative_fig8_330210450

It summarises neatly all the key information an aerobot designer needs with respect to atmospheric conditions vs. altitude. Note the temperature inversion which has implications for the speed of sound vs. altitude.

Section 3.1.2 line 4 – “aspects of this hierarchical classification are shown in Fig. 3.4”

Section 3.1.2 line 5 – “For aerobot design, the fourth-order...”

You state that Fifth-order regional features pose less of a challenge.

I don't quite agree with that statement – they pose *different* challenges – especially to the undercarriage design.

This has to be thought through very carefully to design a system that can robustly sit on the surface correctly, and not be affected by rocks, boulders etc. It also requires the aerobot to have good hazard detection systems such as all-round machine vision, so that it can autonomously land safely (Earth flight control not being possible due to the speed of light delay).

This is not a trivial problem for any large aerobot. NASA's Ingenuity – being very small – was not so badly affected, as it could more easily find the required space between rocks.

Assuming VTOL. Should the undercarriage be stalk-like? 3 or 4 stalks? (3 will always sit on a plane – 4 won't necessarily).

Should pads be used to stop the aerobot sinking into soft terrain – if so, how big?

Should skis be used instead, as for many terrestrial helicopters?

How high should the undercarriage be to avoid rocks under the body?

What are the implications for vehicle stability given the Martian winds?

Should the undercarriage be retractable or not?

What is the effect of dust on such mechanisms?

Should the aerobot use a rocker-arm mobility system similar to that of the rovers? Does it need land mobility? – these are all design choices for the aerobot designer.

Page 9, line 3 – Other distinctive features, the Medusae...

[Latin feature names are usually italicised – *Medusae Fossae*, etc. but check Nature's in-house style.]

Section 3.1.3, line 2 – “...free-air gravity map (120 km resolution (Fig. 3.6) charts these...”

You state that variations in gravity are significant enough to impact the design.

Comparing the force variation to other force variability, e.g. weather, wind loading, flight altitude variation, etc. – just how big a factor is gravity variability – can you please quantify this?

In my view a 1% variation in weight is well within the capabilities of any aircraft control system – indeed the weight may not be known to this precision, given dust loading etc.

A variability of 0.03 N/kg doesn't seem to be much compared, for example, to the aerodynamic forces (drag, lift) and their variability. Consider the weight and drag increase due to dust etc.

The GMM-3 map only has a resolution of 120 km – this seems very large for precise path planning, hazard avoidance, subsurface feature detection etc. you describe.

I think this section needs a more quantitative analysis to support your contention that gravity anomalies are significant. I believe these effects would be “in the noise” given the many other

effects of the Martian environment on the aerodynamic efficiency and control requirements of a specific vehicle and a specific flight path.

Please quantitatively justify these assertions or moderate this section to indicate that the gravity field *may* be a factor that needs consideration if the aircraft is acting at its performance limits or has a particularly sensitive control system. Or simply remove it.

As you state at the end of Section 3.1.2 – the atmosphere (and its physical properties, weather, winds and dust storms, etc.) is a key driver for the aerobot design, and in my view, is *the* dominant one.

Page 12 – surely the weather phenomena associated with the seasonal mass exchanges are more critical than minor temporal gravity variations as far as vehicle aerodynamics are concerned? Winds and seasonal dust storms have the potential for a major impact on the safety and viability of aerobots over the longer term.

Section 3.2.2., para. 1 would be a good place to insert the atmospheric profile graphs mentioned above.

Section 3.2.2.2., line 5 – ...it mobilises...

Section 3.2.2.3, para 2. – here you mention Reynold’s Number which is indeed a key parameter. You might mention that the Reynold’s number for typical Mars aircraft is small compared to the equivalent on Earth – for that reason the flight regime is more reminiscent of insect flight than of traditional terrestrial aircraft design. This requires some re-thinking of design approaches.

Section 3.2.3, line 1 – “...on a daily frequency”

Page 15, line 7 – the word “spacecraft” should be “aircraft” or “Mars aircraft”.

[Yes indeed, as with any spacecraft – charging and ESD mitigation and ionising radiation effects mitigation will be necessary on Mars aerobots.]

Section 3.2.4, line 1. – Mars is subject to significant ionising particle....”

Section 3.2.4 – the supplementary notes S1 add useful detail

Section 3.2.5 – the supplementary notes S2 add useful detail.

Section 4

Section 4 is an excellent and well written summary of the design and testing of the Ingenuity aerobot and the implications for future aerobot design.

Section 4, line 5 – define GPS. (without navigational information provided by systems such as the Global Positioning System (GPS) orbiting Earth.

Section 4, line 8 – “The Ingenuity”

This section also refers to gravity anomalies as being an issue – please see previous comments. I think local weather may be a bigger problem.

Section 4.1.1.1. line 1 “...was conducted in the NASA Jet Propulsion Laboratory (NASA/JPL)...”

Section 5

Section 5 introduces the main contribution of the paper – a methodology for thinking about Mars aerobot design. It is an exemplar of the systems design principles which are (or should be) applied to all space missions. This is an excellent section.

Page 23, para 3, line 6 “2000 RPMs” is a very specific figure and is highly dependent upon the nature of the aerobot. An aerostat, for example, has no need to generate lift this way, of a hybrid aerobot (see the IAC paper referred to above), may have very different requirements.

This should be deleted and replaced by

“...due to the requirement of generating sufficient thrust – and is especially an issue for a helicopter style of aerobot, such as Ingenuity, which has to generate lift aerodynamically via rotor blades which in turn requires rotor tip speeds close to the speed of sound on Mars; therefore, Ingenuity has a flying endurance of only...”

Page 27, para 1, line 7 refers to “aerial spacecrafts” – these are aircraft! A spacecraft that operates in a planetary atmosphere is still an aircraft.

Please note throughout the paper “spacecraft” and “aircraft” should be used for the plural.

Page 27, para 1, line 3 Perseverance is not a deep space spacecraft – its host carrier was but this is a planetary rover – as such it is not operating in space (by definition a vacuum).

RTGs have been used on deep space missions such as Voyager, etc., where the use of solar power is impractical. It has also been used on lunar and planetary surfaces, such as in the Perseverance rover, where a long-term consistent power source, independent of sunlight, has been required.

Section 6

Fine.

Overall, this is a very good paper which introduces the reader to many of the key issues involved in Mars aerobot design – no easy task. The paper provides a good background and a clear exposition of the issues. The use of real and proposed missions from literature illustrates the work very well.

The paper is novel in that it brings all the issues together in one place and offers a structured “systems design” approach to thinking about the design problem.

The paper will be of value to any researcher embarking on Mars aerobot design.

I recommend publication, subject to the corrections outlined in my review.

My comments and mainly regarding issues of English and clarity.

My only significant technical issue comes with the section of the small (1%) variability in gravity measured from orbit over large surface areas – I do not believe this is a significant issue for aerobot design given that there are many other sources of force variability acting on the aircraft.

I remain to be convinced – but that requires a proper comparison of the variability of all the forces and their variability acting on the aerobot.

In the absence of such a quantitative analysis, I would be happy if the section was moderated as I have suggested to say that gravitation variability may need to be taken into account for aircraft acting close to their performance or control limits.

Prof Craig Underwood
Emeritus Professor of Spacecraft Engineering,
Surrey Space Centre,
University of Surrey,
Surrey, GU2 7XH
United Kingdom